# Lateral orbitofrontal cortex anticipates choices and integrates prior with current information

Ramon Nogueira[1,2,*], Juan M. Abolafia[3,*], Jan Drugowitsch[4,5], Emili Balaguer-Ballester[6,7], Maria V. Sanchez-Vives[3,8] & Rubén Moreno-Bote[1,2,9]

Adaptive behavior requires integrating prior with current information to anticipate upcoming events. Brain structures related to this computation should bring relevant signals from the recent past into the present. Here we report that rats can integrate the most recent prior information with sensory information, thereby improving behavior on a perceptual decision-making task with outcome-dependent past trial history. We find that anticipatory signals in the orbitofrontal cortex about upcoming choice increase over time and are even present before stimulus onset. These neuronal signals also represent the stimulus and relevant second-order combinations of past state variables. The encoding of choice, stimulus and second-order past state variables resides, up to movement onset, in overlapping populations. The neuronal representation of choice before stimulus onset and its build-up once the stimulus is presented suggest that orbitofrontal cortex plays a role in transforming immediate prior and stimulus information into choices using a compact state-space representation.

[1] Center for Brain and Cognition and Department of Information and Communications Technologies, Universitat Pompeu Fabra, Barcelona 08018, Spain. [2] Research Unit, Parc Sanitari Sant Joan de Déu, Esplugues de Llobregat, Barcelona 08950, Spain. [3] Institut d'Investigacions Biomèdiques August Pi i Sunyer (IDIBAPS), Barcelona 08036, Spain. [4] Département des Neurosciences Fondamentales, Université de Genève, Geneva 4 1211, Switzerland. [5] Department of Neurobiology, Harvard Medical School, Boston, Massachusetts 02115, USA. [6] Department of Computing and Informatics, Faculty of Science and Technology, Bournemouth University, Poole BH12 5BB, UK. [7] Bernstein Center for Computational Neuroscience, Central Institute of Mental Health, Medical Faculty Mannheim/Heidelberg University, Mannheim D-68159, Germany. [8] ICREA, Barcelona 08010, Spain. [9] Serra Húnter Fellow Programme, Universitat Pompeu Fabra, Barcelona 08018, Spain. * These authors contributed equally to this work. Correspondence and requests for materials should be addressed to R.M.-B. (email: ruben.moreno@upf.edu).

Making a decision in real life requires the integration of preceding and current information to adaptively guide behavior[1,2]. Previous work has investigated the neuronal regions responsible for achieving this goal by using experimental paradigms where the sequence of external events, or history, flows independently of the choices of the actor[1,3,4]. In many cases, however, choices of an actor can influence future external events, and so to speak, change the course of history. Relatively less work has been devoted to the study of tasks in which recent past information matters for the current choice and immediately previous choices affect the upcoming states of the world[2,5–8].

The orbitofrontal cortex (OFC), like other regions in the prefrontal cortex, is thought to play an important role in adaptive and goal-directed behavior[9–15]. Previous single-neuron accounts have demonstrated that OFC encodes a myriad of variables that are relevant for behavior in decision-making[12], such as primary rewards and secondary cues that predict them[16,17], values of offered and chosen goods[18–20], choices and responses[19,21–24], expected outcomes[25] and stimulus type[26], while human brain imaging studies have corroborated and largely extended these results[9,27–30]. However, in contrast to other prefrontal and parietal brain areas[3,31,32], the OFC displays relatively weak choice-related signals[19,22–24]. Further, neuronal signals anticipating upcoming choices before stimulus onset have not been described, except in a single report in monkeys[22]. This has led to the predominant view that OFC is not responsible for action initiation and selection[14,20,21]. Here, in contrast, we hypothesize that OFC plays a central role in decision-making, first, by representing the central latent variables of the task (state-space) and, second, by combining the most recent past with current stimulus information. We hypothesize also that this combination of information happens through a compact representation of the task's state-space, that is, by representing predominantly the variables of the immediate past that are critical to perform the task. We support this hypothesis through our findings that OFC (1) represents choice initiation and choice selection even before sensory evidence is available, (2) encodes the state-space determined by just the previous trial (here called immediate prior or immediate past information), (3) integrates the immediate prior information with current sensory evidence and (4) promotes filtering out behaviorally irrelevant variables.

In this study we use an outcome-coupled perceptual decision-making task that requires integrating prior information from the previous trial with an ambiguous stimulus. This task is designed to maximize the chances of revealing choice initiation and choice selection signals that integrate both immediate prior and current information. Rats efficiently solve this task by using the relevant second-order combination of previous choice and reward and combining this most recent prior information with currently available information of a perceptually challenging stimulus. On the basis of single-neurons and simultaneously recorded neuronal ensembles in the lateral OFC (lOFC), we find a build-up of choice-related signals across time; critically, upcoming choice can be traced back to a period of time before stimulus onset. Overlapping neuronal populations encode choice, immediate prior and stimulus information stably over time up to movement onset. These neuronal populations represent behaviorally relevant variables in a task-structure dependent way. For example, information about the immediate past cease to be represented once such variables become behaviorally irrelevant due to a change in the task structure. Similarly, in the main task, the coexistence of choice-related and latent variables within the same neuronal circuits enables lOFC to play an important role in integrating prior with stimulus information to aid choice formation using a compact state-space representation. Our results are consistent with the hypotheses that OFC plays a role in the temporal credit-assignment problem, the problem of correctly associating an action with a reward delayed in time[9,14] and in representing latent states[11]. Furthermore, our work adds the view that lOFC might play a central role in decision-making by integrating immediate prior information with current information through a refined encoding of the state-space in the task.

## Results

**Animals use task-contingencies to improve performance.** Rats performed a perceptual decision-making task (Fig. 1a), which in each trial consisted in classifying an inter-tone time interval (ITI), as short ($S = s$) or long ($S = l$). The rats self-initiated the trial with a nose poke in the central socket, after which they had to hold the position until the ITI had completely elapsed. A correct response was defined as poking into the left socket if the stimulus was short, and into the right socket if the stimulus was long, after which the rat was rewarded with water. A stimulus was considered difficult if the inter-tone interval was close to the category boundary, and easy otherwise (Fig. 1a). Importantly, in our task the choices of the animal influenced the history of future events. Specifically, in the trial following a correct response ($R = +1$), the ITI was drawn uniformly at random from eight possible values, while in trials following an incorrect response ($R = -1$), the stimulus was repeated (Fig. 1b). This sequence created a rich environment, whereby in many trials the ITIs were not drawn randomly. Rather, the environment was formally described as an outcome-coupled hidden Markov chain, that is, a Markov chain in which the sequence of trials is coupled with the outcomes of the animals' choices. The Markov chain was hidden because of two reasons (Supplementary Fig. 1): first, due to potential limits in memory and attention, we did not consider previous trials as fully known; and second, the stimulus was not fully visible at any trial, especially so in the most difficult trials (Fig. 1a). The combination of independent trials after correct responses and fully dependent trials after incorrect responses allowed us to distinguish signals from the past from those that anticipated upcoming events, as discussed in the next section.

From an ideal observer's perspective, there is critical information that the animal should monitor to perform the task efficiently. The outcome in the previous trial, $R_{-1}$, determines whether the stimulus in the next trial will be repeated or drawn randomly: if the previous outcome was incorrect ($R_{-1} = -1$), then the stimulus will be repeated in the next trial, while if the previous trial was correct ($R_{-1} = +1$), then the next stimulus will be randomly drawn. Therefore, if the animal tracks the outcome $R_{-1}$, its behavior will improve because it could often anticipate the stimulus. In fact, the three rats learnt this task contingency by using the previous outcome to improve their behavior (Fig. 1c; individual rats and fits shown in Supplementary Fig. 2). First, all animals featured a psychometric curve (computed after correct trials) with a larger fraction of correct responses for easy than for difficult trials (rat 1: difference = 9.8 pp (percentage points), non-parametric one-tailed bootstrap, $P < 10^{-4}$; rat 2: difference = 10 pp, $P < 10^{-4}$; rat 3: difference = 8.0 pp, $P < 10^{-4}$; see Methods). Importantly, when the psychometric curve was computed after incorrect trials, the slope of this curve increased significantly for all rats (rat 1: percentage change = 42%, non-parametric one-tailed bootstrap, $P = 4.4 \times 10^{-3}$; rat 2: percentage change = 81%, $P < 10^{-4}$; rat 3: percentage change = 110%, $P = 5 \times 10^{-4}$). The improvement was substantial, with an average relative increase of 9 pp in performance in difficult trials after incorrect responses compared to after correct responses (non-parametric one-tailed bootstrap, $P < 10^{-4}$).

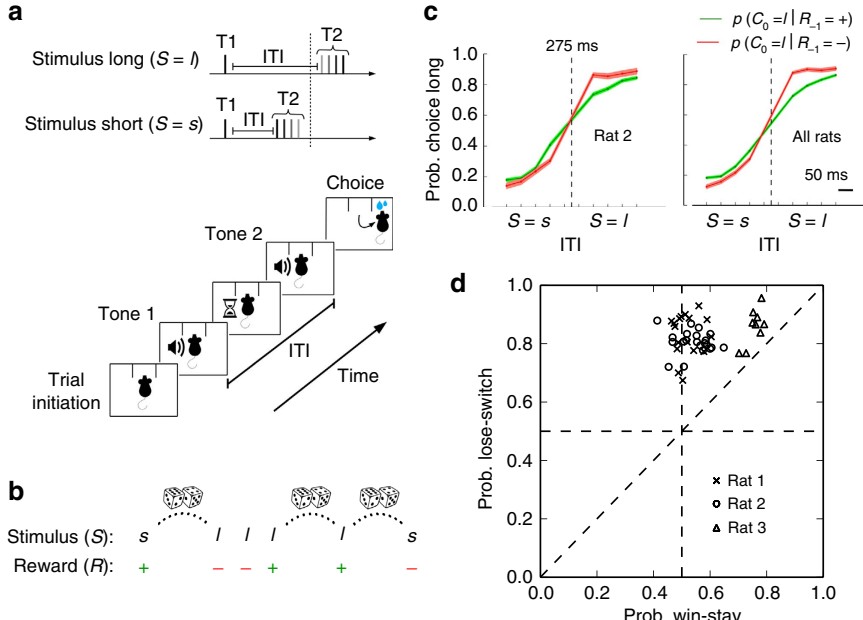

**Figure 1 | Rats use the trial-by-trial-dependent contingencies of the task to improve their performance.** (**a**) Schematic of the task (see Methods for details). Two identical, consecutive tones (T1 and T2) are presented to the rats (top panel). Inter-tone intervals (ITIs) can belong to two stimulus categories: short, $S = s$, or long, $S = l$. Each category has four possible ITIs (short: 50, 100, 150 and 200 ms; long: 350, 400, 450 and 500 ms). The vertical dotted line represents the decision boundary at 275 ms. Difficult ITIs, depicted in gray, lie close to the decision boundary. Sequence of events within a trial (bottom panel): from trial initiation to choice. Rats self-initiate the trial and sample the stimuli in the central socket. They are rewarded with water if they poke the right socket when the stimulus is long, and the left socket when the stimulus is short (for rat 3 the contingency was the opposite). (**b**) The sequence of trials follows an outcome-coupled hidden Markov chain (see also Supplementary Fig. 1): a new random stimulus condition is presented after a correct response ($+$), while the same stimulus condition is presented after an incorrect response ($-$). (**c**) Psychometric curves (probability choice long versus ITI) after correct responses (green line) and after incorrect responses (red line) for an example rat (left panel) and for all rats (right). The slope of the psychometric curves after incorrect responses substantially and significantly increases relative to the slope of the curve after a correct response. Error bars (shaded) are estimated by bootstrap (one s.d.). (**d**) Probability of lose-switch versus probability of win-stay. Each point corresponds to a different session. Rats predominantly follow a lose-switch over a win-stay strategy. No strategy being followed corresponds to the point (0.5, 0.5) in the plot.

Consistent with the observation that the animals use the structure of the outcome-coupled hidden Markov chain to improve their behavior, we also found that on a session by session basis animals predominantly followed the lose-switch part of a win-stay-lose-switch strategy with a substantially weaker win-stay part (Fig. 1d; all rats: difference lose-switch—win-stay probabilities $= 0.24$ pp; non-parametric one-tailed bootstrap, $P < 10^{-4}$; see Methods). Following a lose-switch strategy with no win-stay bias would lead to optimal behavior in our task if, ideally, the Markov chain were fully visible (not hidden). However, the actual ITI category in each trial is unobserved (because some trials are difficult) and the past might not be fully known due to memory leak. Consistent with this, the rats displayed some departures from the optimal strategy, in particular featuring a significant win-stay component in their behavior (rat 1: mean 0.51, non-parametric one-tailed bootstrap, $P = 1.0 \times 10^{-3}$; rat 2: mean $= 0.54$, $P < 10^{-4}$; rat 3: mean $= 0.75$, $P < 10^{-4}$).

The observed changes in the psychometric curve suggest that animals track a variable that jointly monitors previous choice $C_{-1}$ ($C_{-1} = -1$ if the choice was long, or $C_{-1} = +1$ if it was short) and previous outcome $R_{-1}$. This second-order prior variable informs the rat about what choice it should make after an incorrect response, and mathematically is expressed as $X_{-1} = C_{-1} \times R_{-1}$ (Methods). The state-space in our task consists both of the previous outcome and second-order prior, because these two variables fully define all that needs to be known by the rat to behave efficiently in this task. These two variables also fully define the prior information that is task-relevant, called

immediate prior information. To confirm the prediction that the rats keep track of the second-order prior, $X_{-1}$, we asked how well past events are able to predict the upcoming choice $C_0$. Among the large number of behavioral variables that could influence upcoming choices, we found that the second-order prior $X_{-1}$ was the most predictive quantity, only surpassed by the stimulus itself, $S_0$ and followed by the previous outcome $R_{-1}$ (Supplementary Fig. 3; Supplementary Methods).

**Single-cells encode upcoming choice and second-order prior.** We looked for neural coding of immediate prior information and upcoming choices throughout the trial. Tetrodes were inserted in the right hemisphere of the rat lOFC (Fig. 2a). Small ensembles of well-isolated single units were simultaneously recorded (mean size $= 2.9 \pm 1.6$ neurons). Our dataset consisted of a total of 137 single-neurons with an average of 684 behavioral trials, eliciting a median of 9000 spikes per neuron, before excluding neurons with mean firing rates below 1 Hz (including all cells did not qualitatively influence the results; for a detailed description of the total number of cells for each analysis see Methods and for an additional power analysis for the number of cells and rats see Supplementary Methods). Recordings started after rats had reached a performance of at least 75%.

Our behavioral results suggest that the animals closely monitor second-order prior, $X_{-1}$, and other variables that correlate with it, such as previous choice $C_{-1}$ and resulting outcome $R_{-1}$. We reasoned that if OFC participates in the decision-making process, then OFC neurons should encode these variables as well as reveal signals that anticipate upcoming choices. To test this

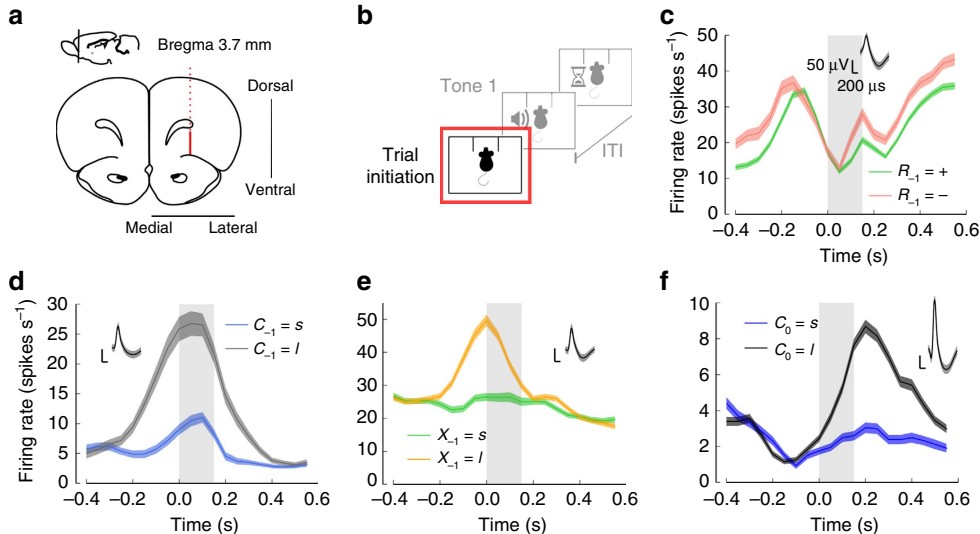

**Figure 2 | OFC neurons encode relevant past information and anticipate upcoming choices even before stimulus onset. (a)** Electrode's path (dashed red line) and recording sites (solid red line) in rat lOFC depicted in a coronal section representation at 3.7 mm AP, 2.5 mm ML and 1.6 mm DV from Bregma. **(b)** Neuronal responses were aligned to trial initiation, defined as the time at which the rat starts the trial by poking the central socket. **(c)** Example neuron encoding reward in the previous trial $R_{-1}$ (either $+$ or $-$). This particular neuron fires more strongly for non-rewarded previous trials. **(d)** Example neuron representing choice in the previous trial $C_{-1}$ (either $s$ or $l$). **(e)** Example neuron tracking second-order prior $X_{-1} = C_{-1} \times R_{-1}$ (either $s$ or $l$). **(f)** Example neuron encoding upcoming choice $C_0$ as a function of time. This neuron conveys information about rat's upcoming choice before stimulus onset (stimulus onset always happens to the right of the shaded area). **(c–f):** Time zero corresponds to trial initiation. The period of time between trial initiation and the shorter stimulus onset (150 ms) is indicated with shaded areas. Curves correspond to trial-averaged firing rates smoothed with a causal sliding rectangular window (size of 100 ms and step of 50 ms), and shaded areas around them correspond to s.e.m. Insets represent spike waveform for each neuron (black line, mean; shaded area, s.d.).

prediction, we initially focused on the trial initiation period, where the stimulus has not yet been presented. We first aligned the neuronal responses to the initiation of the trial (Fig. 2b). Before performing pooled population-level analyses, we will first focus on the tuning of some example neurons. We found neurons whose trial-averaged activity illustrated a diversity of behaviors associated with both backward and also forward events. In Fig. 2 we show some individual examples. We identified neurons that showed conspicuous modulations as a function of the previous outcome (Fig. 2c), previous choice (Fig. 2d), second-order prior (Fig. 2e) and interestingly, also about upcoming choice (Fig. 2f). The neuron shown in Fig. 2f could predict upcoming choice with an accuracy of 71% (AUC, see Methods).

These quantities were also encoded throughout the trial (Fig. 3). Just before stimulus offset (Fig. 3a–d), when the animal is still poking into the central port, stimulus information is strongly represented in some neurons in lOFC (Fig. 3b). Signals about the upcoming choice were also clearly visible in this pre-movement period (Fig. 3c). This neuron predicted upcoming choice with 84% accuracy (AUC). Finally, the firing rate of some cells was modulated by the expected value of the outcome, $EV_0$ (Fig. 3d; Methods). When we analysed single-neuron responses at lateral nose poking onset (Fig. 3e), we found neurons whose rate was largely modulated by stimulus (Fig. 3f). Signals about the current choice were also strongly present, as shown by the example neuron in Fig. 3g. This neuron predicted the performed choice with 87% accuracy (AUC). We also observed outcome-modulated neurons in this period (Fig. 3h). Thus, even single-neuron activity by itself already provided strong indication that lOFC was representing the task-relevant variables.

**OFC encodes immediate prior and anticipates future choices.** We confirmed the single-neuron observations at the population level with a Generalized Linear Model (GLM) analysis of the spike count responses of single-neurons. To do so, we regressed

the spike count of each single-neuron simultaneously against a large set of variables, including the stimulus, reward, choice, difficulty and second-order prior of the current trial, the previous trial and up to three trials in back (Methods). This approach was preferred over a receiver operating characteristic (ROC) approach because the latter might find significant AUC values even in the absence of veridical encoding of the variable, simply due to correlations with other encoded variables (see Methods).

Before stimulus onset, we found that a significant fraction of neurons (25%, one-tailed binomial test, $n = 76$, $P = 4.6 \times 10^{-9}$) predicted the upcoming choice, $C_0$ (Fig. 4a). Significant fractions of cells also encoded the second-order prior $X_{-1}$, previous choice $C_{-1}$, and the previous outcome $R_{-1}$. Thus, the neurons shown in Fig. 2 represent just examples of potentially overlapping large neuronal populations that encode these variables. Interestingly, we did not find a substantial fraction of cells encoding information from two or more trials into the past, suggesting that information older than arising from the preceding trial is not present in lOFC.

We found that cells encoded both current stimulus $S_0$ and current outcome $R_0$ ($S_0$ and $R_0$, 11% each, one-tailed binomial test, $n = 76$, $P = 0.036$) even before stimulus onset. Although at first glance surprising, this result arises from the outcome-coupled hidden Markov chain structure of the environment. In fact, when we repeated our GLM analysis using only trials after correct responses—where the upcoming stimulus cannot be predicted from the stimulus used in the previous trial—we found that neither stimulus $S_0$ (9%, one-tailed binomial test, $n = 76$, $P = 0.085$) nor reward $R_0$ (9%, one-tailed binomial test, $n = 76$, $P = 0.085$) information was present before the onset of the stimulus (Supplementary Fig. 4). Focusing instead only on trials after incorrect responses, we again found that a substantial fraction of cells (14%, one-tailed binomial test, $n = 76$, $P = 1.3 \times 10^{-3}$) can predict the stimulus. Altogether, our results show that, before stimulus onset, lOFC tracks the second-order

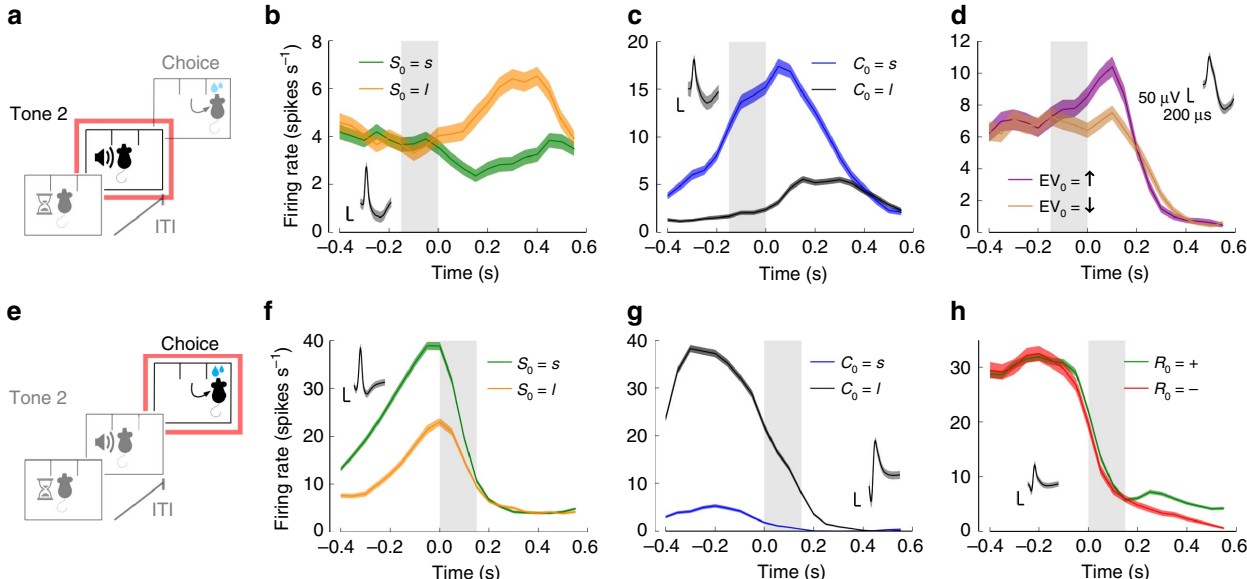

**Figure 3 | OFC neurons encode essential quantities throughout the trial.** (**a**–**d**) Neuronal responses were aligned to stimulus offset (**a**). The firing rate of OFC neurons was modulated in a time period before stimulus offset (150 ms, shaded areas in **b**–**d**) by the stimulus (**b**), the upcoming choice (**c**) and the expected value of the outcome (**d**). (**e**–**h**) Neuronal responses were aligned to lateral nose poking onset, choice period (150 ms) (**e**). The firing rate of OFC neurons represented the stimulus (**f**), the current choice (**g**) and the outcome (**h**). In the two periods, signals about upcoming and current choice were very conspicuous. Time zero corresponds to stimulus offset (**b**–**d**) and lateral nose poking onset (**f**–**h**). Curves correspond to trial-averaged firing rates smoothed with a causal sliding rectangular window (size of 100 ms and step of 50 ms), and shaded areas around them correspond to s.e.m. Insets represent spike waveform for each neuron (black line, mean; shaded area, s.d.).

prior $X_{-1}$, and anticipates the upcoming choice, $C_0$. Thus, rat lOFC carries sufficient information to play an important role in integrating immediate prior information with sensory information.

**Build-up of choice-related neuronal signals**. If OFC represents the integration of immediate prior with current information, then information about upcoming choices should increase as further evidence is integrated into the system. For instance, just before stimulus offset, information about the stimulus is readily available, and should be combined with prior information to inform decisions. In fact, a substantial fraction of cells encoded the upcoming choice $C_0$ just before stimulus offset (Fig. 4b). This fraction was large (30%, one-tailed binomial test, $n = 87$, $P = 7.6 \times 10^{-14}$), and larger than during the pre-stimulus period (see Fig. 4a,b; difference = 5 pp, one-tailed non-parametric difference binomial test, $P = 0.25$; see Methods). Integration of information at the population level could be accomplished within the same circuit, as a large fraction of cells also encoded the stimulus $S_0$ in the current trial (33%, one-tailed binomial test, $n = 87$, $P = 1.1 \times 10^{-16}$). Interestingly, in the choice period, 77% of all cells (60/78 neurons) encoded choice (Fig. 4c) –significantly more than in the pre-stimulus periods (Fig. 4a,b; difference = 52 pp, one-tailed non-parametric difference binomial test, $P < 10^{-4}$). Thus, there is a build-up of choice-related signals in lOFC, as illustrated when plotted as a function of the analysis time period (Fig. 4d).

Stimulus also seemed to be encoded in OFC in a sensible way, with information peaking before stimulus offset. We found that the fraction of neurons encoding stimulus $S_0$ increases significantly from trial initiation to the stimulus offset period (Fig. 4d; difference = 23 pp, one-tailed non-parametric difference binomial test, $P = 2.2 \times 10^{-4}$) and decreases significantly thereafter (difference = 18 pp, one-tailed non-parametric difference binomial test, $P = 3.9 \times 10^{-3}$). Encoding of past task events, such as second-order prior, previous choice and previous reward, declined as time progressed over the trial (Fig. 4d; $C_{-1}$:

difference = 12 pp, one-tailed non-parametric difference binomial test, $P = 0.036$; $X_{-1}$: difference = 11 pp, $P = 0.033$; $R_{-1}$: difference = 19 pp, $P = 2.2 \times 10^{-3}$; differences computed between pre-stimulus and choice periods). Altogether, these time profiles suggest that information about stimulus and second-order prior is incorporated into choice-related signals to mediate the integration of information.

We found a correlation between the encoding weights for upcoming choice computed at the pre-stimulus and stimulus offset periods (Fig. 5a; Methods). The same was observed for the weights computed for second-order prior, previous choice and previous reward. This suggests that the encoding of these variables is partially sub-served by stable populations during the periods of time in which prior information needs to be integrated with sensory information. However, their encoding differed in the choice period, precisely when sensory information does not need to be integrated any more, as not such correlation was observed (Fig. 5b). In particular, the increase of choice-encoding neurons over time reported in Fig. 4d suggests that the lack of correlation between encoding during stimulus offset and choice periods might arise from a recruitment of additional choice-related cells, potentially motor-related. We also found that the encoding weights of second-order prior and upcoming choice were positively correlated during the pre-stimulus period (Fig. 5c; Methods), suggesting that populations of neurons encoding the previous trial's state and upcoming choice partially overlap before stimulus presentation.

Some differences in behavior across animals were clear (see Fig. 1d and Supplementary Fig. 2), with rat 3, for instance, displaying a higher lose-switch probability than the other rats. We first confirmed in a separate analysis that none of the qualitative results described above changed when neurons recorded from this rat were excluded from the analysis. We also confirmed that rat-by-rat analysis of neuronal populations delivered the same trends as reported above, generally including encoding of upcoming choice before stimulus onset and the ramping of

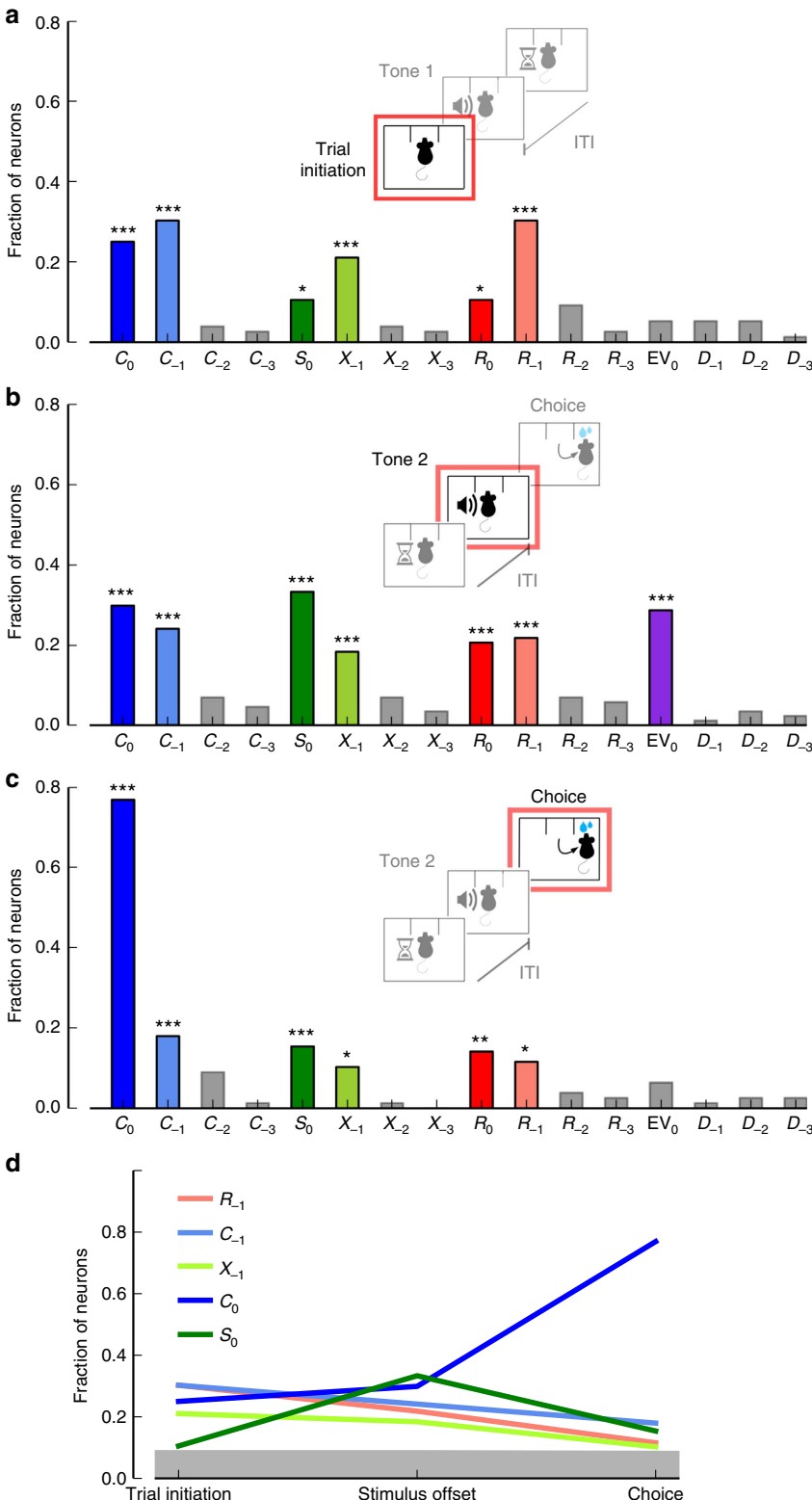

**Figure 4 | Neurons in lOFC integrate prior with current sensory information and encode upcoming choice.** (**a**) Fraction of neurons with significant regressors (see Methods) for each of the variables listed in the horizontal axis. Upcoming choice $C_0$, previous choice $C_{-1}$, upcoming stimulus $S_0$, second-order prior $X_{-1}$, upcoming outcome $R_0$ and previous outcome $R_{-1}$ are significantly encoded in the population. Upcoming choice $C_0$ is encoded by lOFC neurons even before stimulus is presented. Variables that extend further back into the past are not significantly encoded in lOFC. (**b–c**) Fractions of neurons with significant regressors during a time period before stimulus offset (**b**), and during a time period after lateral nose poking (**c**). (**d**) Fractions of neurons encoding upcoming choice $C_0$, stimulus $S_0$, second-order prior $X_{-1}$, previous choice $C_{-1}$ and previous reward $R_{-1}$ at trial initiation (pre-stimulus), stimulus offset and choice periods. Note that larger fractions of neurons have choice-related signals as time progresses through the trial. (**a–c**) One-tailed binomial test, $* = P < 0.05$, $** = P < 0.01$, $*** = P < 0.001$. Shaded rectangle corresponds to non-significant fraction of neurons ($P > 0.05$).

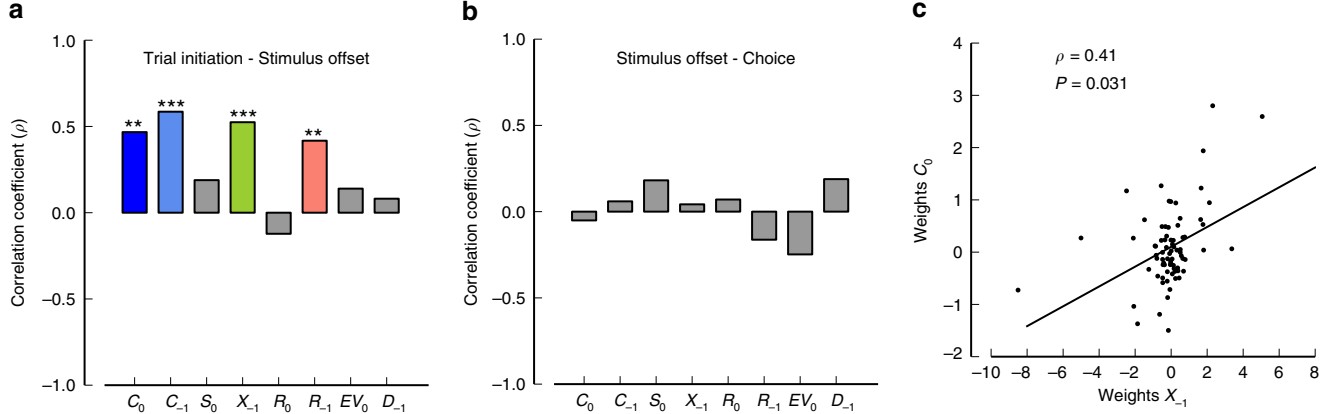

**Figure 5 | Encoding of essential variables for the task is stable before motor execution of the choice.** (**a**) Correlation coefficient between weights estimated at trial initiation and before stimulus offset for several variables (see Methods). Upcoming choice $C_0$, second-order prior $X_{-1}$, previous choice $C_{-1}$ and previous reward $R_{-1}$ are stably encoded in lOFC. (**b**) None of the correlation coefficients between weights computed just before stimulus offset and during the choice period were significantly different from zero. (**c**) There is a positive correlation between the encoding weights associated with second-order prior and upcoming choices at the pre-stimulus period. Two-tailed permutation test (see Methods), $* = P < 0.05$, $** = P < 0.01$, $*** = P < 0.001$.

choice-related information across time periods (Supplementary Fig. 5).

**Expected value and outcome representations.** After stimulus presentation, at stimulus offset, the animal might have a sense of how difficult the trial was. This informs about the subjective probability (confidence) of getting a reward, as easy trials should promise a more secure reward than difficult trials. Since in our experimental setup we do not vary the reward amount, encoding the subjective probability of a positive outcome amounts to the expected value in the current trial, which in turn is inversely related to the difficulty of the trial (see Methods). In this time epoch, the expected value was encoded in a large fraction of cells (Fig. 4b; 29%, one-tailed binomial test, $n = 87$, $P = 6.1 \times 10^{-13}$). Previous work has also found that signals about decision confidence are encoded in the activity of single-cells in rat OFC (ref. 33), and in monkey parietal cortex[34]. We also found in this period of time a large fraction of cells that encode outcome in a predictive way, as this variable can be partially inferred based on the difficulty of the trial. Outcome was also encoded at the choice period (Fig. 4c), consistent with the role of this area in encoding reward and outcomes[16,17].

**Behaviorally irrelevant prior is not represented in OFC.** The previous results demonstrate that OFC represents state-space when rats are in an environment where it is behaviorally advantageous to keep track of this information. We tested the encoding of immediate prior information when this information was irrelevant by placing the same rats in an environment where they were passively exposed to the same set of stimuli but rewards were not delivered. Rats were exposed to two passive stages, before and after the decision-making stage (see Methods). We found that OFC did no longer keep track of the immediate prior information (defined as previous stimulus $S_{-1}$ in the passive environment, equivalent to $X_{-1}$ in the decision-making stage; see Methods) at any time during the trial (Supplementary Fig. 6). Encoding of current stimulus and difficulty at the stimulus-offset period weakly persisted in this environment, suggesting that task-irrelevant variables observable at the current trial are not completely filtered out in OFC. These results suggest that OFC does not monitor state-space from the immediate past when this information is task-irrelevant.

**Population decoding reveals a hierarchy of variables in OFC.** Our previous analysis has revealed that, following correct choices, only two variables are significantly encoded in the pre-stimulus period in single OFC neurons, namely, second-order prior and upcoming choice (Supplementary Fig. 4). We confirmed that this result holds using a much more stringent test that does not assume that both variables are encoded linearly, as we did before. To do so, we used decoding techniques that predict one quantity at a time from the population activity of a simultaneously recorded neuronal ensemble[35], while keeping the other quantity constant (Fig. 6; Methods). We found that a classifier trained on the pre-stimulus activity of a neuronal ensemble at fixed second-order prior $X_{-1}$ conveyed substantial information about upcoming choice (Fig. 6a). Similarly, when conditioning the activity to upcoming choice $C_0$, we found that small neuronal populations conveyed substantial information about second-order prior (Fig. 6b). These results hold both across all neuronal ensembles in the dataset and when selecting only the 10% most informative ensembles. Decoding performance increased monotonically with the number of neurons in the ensemble (Fig. 6a,b)[36,37]. Because this conditioning-based decoding analysis does not assume that these two variables are both encoded linearly, in contrast to our previous analysis (Fig. 4), these results add strong support to the conclusion that both immediate prior information (that is, second-order prior) and upcoming choice are encoded in lOFC.

Which variables are most readily decoded at the population level? The analysis from the previous sections would suggest upcoming choice and prior information as strong contenders. However, this analysis was based on single neurons and ignored correlations that might be present in neuronal populations and might influence the representation of those variables. To more directly address this question, we trained a classifier as in the previous paragraph to decode per trial individual variables from the activity of small neuronal ensembles (Methods). Using this approach, we found that, consistent with the previous linear encoding analysis (Fig. 4d), the 10% most informative neuronal ensembles had larger amounts of information about upcoming choice than about any other variable (Fig. 7) from the pre-stimulus to the choice periods. Information about the upcoming choice $C_0$ was so strongly present in lOFC that it could be predicted from holdout data not used to train the classifier with an accuracy of 57% for all ensembles and 76% for

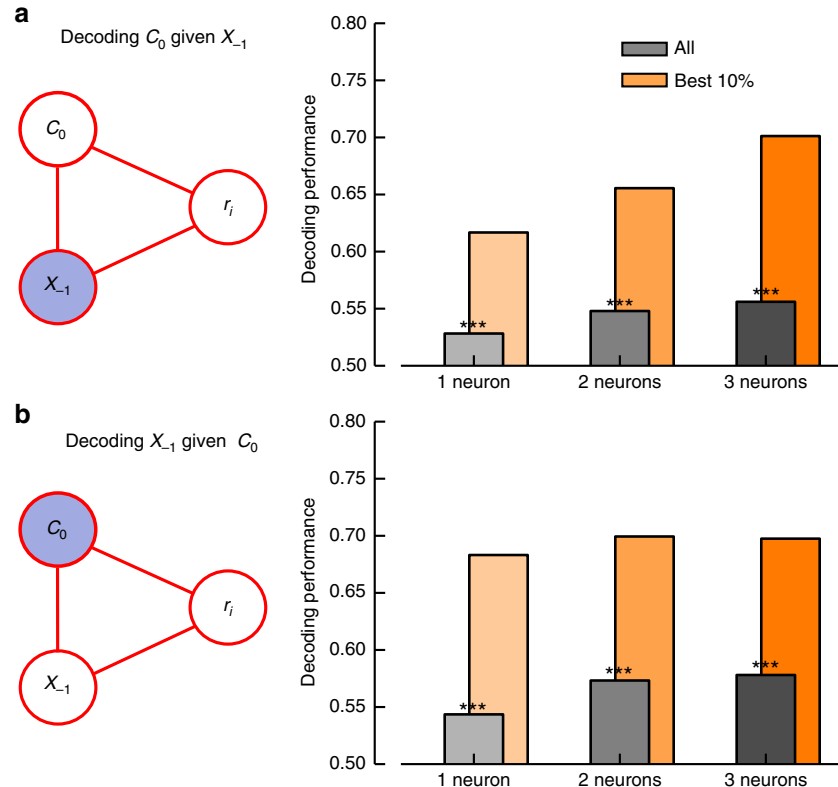

**Figure 6 | Population decoding reveals pre-stimulus neural representations of second-order prior and upcoming choice. (a)** Decoding performance for upcoming choice at fixed second-order prior increases with the number of neurons in the ensemble (one to three) across all ensembles (gray), and does so more strongly for the 10% most informative ensembles (orange). Only trials after correct responses are used for the analysis. Left panel: schematic showing that the pre-stimulus firing rate of neuron $i$ in the ensemble can possibly depend at most on second-order prior $X_{-1}$ and upcoming choice $C_0$, as previously revealed by a linear analysis (Supplementary Fig. 4). To show that upcoming choice truly modulates neural activity, we performed a conditioned analysis by which the value of the second-order prior is fixed (gray-blue) while a linear classifier is trained to predict upcoming choice from the activity patterns in OFC (see Methods). **(b)** Decoding performance for second-order prior at fixed upcoming choice. Colour code and analysis are as in the previous panel. One-tailed permutation test, $* = P < 0.05$, $** = P < 0.01$, $*** = P < 0.001$.

the top 10% ensembles in the pre-stimulus period, 64 and 78% at the stimulus offset period, and 79 and 92% at the choice period (Fig. 7a–c), respectively. The population decoding analysis also again revealed second-order prior as one of the most prominently encoded variables (Fig. 7a–c). Other variables were also decodable from the lOFC, but less accurately. Therefore, the population decoding analysis confirms that lOFC tracks prior information on a trial by trial basis and predicts upcoming choice.

Finally, in view of the individual behavioral differences across animals, we sought to determine whether they were correlated with neuronal differences. We found a positive correlation between lose-switch probability and neuronal information about both upcoming choice and second-order prior, although this correlation did not reach significance (Supplementary Fig. 7; permutation test, $n = 3$, $P = 0.16$, Supplementary Methods). Thus, animals that were more likely to switch after an incorrect response tended to provide a better information-readout in OFC ensembles about variables that are strongly linked to that switching behavior.

## Discussion

OFC is thought to play an important role in adaptive and goal-directed behavior[9–15]. However, as OFC has been shown to encode a myriad of variables, including outcomes, expected rewards and values[12,16–25], a coherent picture of its function is still missing. Previous work on reversal learning[38–40] and Pavlovian-instrumental transfer[41] has revealed that OFC function

reflects crucial aspects of learning, particularly by developing novel representations of associations between cues and their predicted rewards[40,42,43], and by tracking the history of previous outcomes and choices during reward-guided decisions[44,45]. These results show that OFC is important to process prior information that builds over an extended sequence of previous trials to guide behavior. However, it is not well known whether this goal is accomplished through a compact representation of the task's state-space, or by representing all sorts of task-relevant and task-irrelevant variables. Further, whether state variables can be represented exclusively from the previous trial at a high temporal resolution is not known.

We specifically tackled these questions by using a novel perceptual decision-making task endowed with an outcome-coupled hidden Markov chain. By introducing outcome-dependent correlations between consecutive stimuli, we ensured that the animal needed to track on a trial by trial basis the most recent past information to solve the task efficiently. This experimental design maximized the chances of finding state variables that need to be represented at high temporal resolution. It also maximized the chances of identifying interactions of these variables with choice-related signals during the decision-making process. In addition, by inserting random trials after correct responses, an analysis based on systematically conditioning on different task variables allowed us to distinguish neuronal signals that were purely associated with either the immediate past (for example, second-order prior) or future (upcoming choice) events.

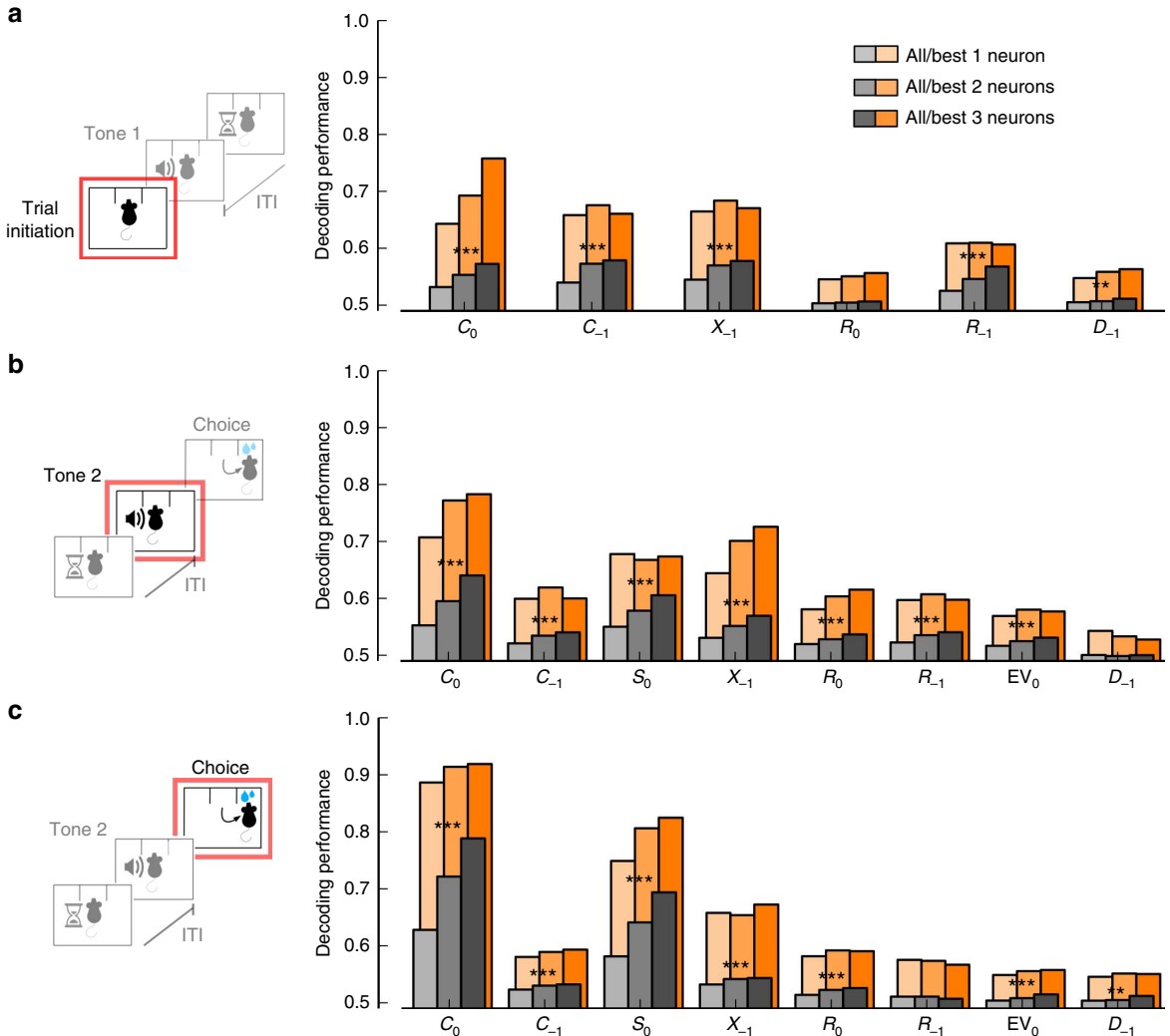

**Figure 7 | Population decoding analysis reveals a hierarchy of encoded variables. (a)** Decoding performance for each quantity at the pre-stimulus period as a function of the number of neurons in the ensemble (one to three) across all ensembles (gray) and for the 10% most informative ensembles (orange). All trials are used for the analysis. **(b,c)** Same as in the previous panel for stimulus offset and choice periods. This analysis reveals a hierarchy of encoding, with upcoming choice $C_0$ and second-order prior $X_{-1}$ being two of the most strongly encoded variables. One-tailed permutation test, $* = P < 0.05$, $** = P < 0.01$, $*** = P < 0.001$.

Thus, this task constitutes an important contribution to the classical perceptual decision-making literature by adding the necessity of considering immediate prior information. Indeed, except for some notable exceptions[2,5,8,46], the study of perceptual decision-making has been dominated by paradigms where sensory information, presented in a random sequence of trials, suffices to inform a correct choice such that prior information from the previous trial can and should be ignored altogether[1,47]. In this line, many studies have emphasized continuous integration of information over time within a trial[1,3,48]. As a consequence, relatively less work has focused on the discrete-like process required to integrate proximal prior events with sensory information[49].

One important feature of our task is that relevant prior information was exclusively present in the previous trial. This immediate prior information was encapsulated in the second-order prior variable $X_{-1}$, the interaction between previous trial choice and reward. The second-order prior along with the previous outcome fully defined the state-space in our task. Our results show that lOFC represents the structure of the task in a compact way, as we found that second-order prior was

among the most strongly encoded variables in lOFC. Our results are in line with a theoretical proposal[11] recently supported by human functional magnetic resonance imaging (fMRI) and rat inactivation studies[46,50] that OFC represents the state-space, and hence add electrophysiological single-cell and neuronal population evidence for such theoretical scenario. In contrast, previous work has shown that in other brain areas, like the dorsolateral prefrontal cortex in monkeys, both task-relevant and task-irrelevant information is encoded in value-based decision-making[47,51]. In addition, we also embedded animals in an environment in which they had to ignore prior information. In this environment, immediate prior information seemed to be abolished in OFC, suggesting that OFC differentially represents state variables that are relevant for the task.

Another important question is the degree of involvement of OFC in the decision-making process. We found a definite encoding of choice-related variables throughout the decision process, appearing even before stimulus onset. This result is consistent with recent work where monkey OFC population activity has been postulated to represent an internal deliberation mediating the choice between two options[52]. It is also in line with

a large body of work showing that OFC plays an important role in goal-directed behavior and thus in action initiation and selection (for example, see refs 46,53–57). Previous work has also found evidence that a multitude of areas are involved in action initiation and selection, such as parietal and prefrontal areas[3,4,19,21,24,31]. However, our results constitute the first report of the existence of neurons in the rodent OFC that have predictive power about upcoming choices before stimulus onset. Interestingly, some of these neurons were found to anticipate upcoming choice with a success probability of 69% (out of 750 test trials not used for training, 520 were correctly predicted using logistic regression), thus demonstrating the presence of strong choice-encoding neurons in OFC even during the pre-stimulus period. At the population level the fraction of neurons encoding for upcoming choice before stimulus onset was strong and highly significant.

Finally, we found evidence that the observed compact representation of state-space in OFC can play a role in integrating immediate prior with current information. First, we found a strong representation of current stimulus information that declined after stimulus offset, an effect that was accompanied by a large increase of choice-related signals representing the integration of stimulus with prior information. This result suggests that the neuronal representation of the state-space interacts in the OFC with the decision-making process, potentially by facilitating the combination of prior with current information. This result is consistent with a recent human fMRI study suggesting that OFC represents posterior probability distributions by integrating extended prior experience with current information[58].

All in all, our results provide an integrative view of the rodent lOFC by showing that it predominantly represents state-space (in particular, second-order prior), the integration of immediate past with current information, and the initiation and selection of choices. Our results, finally, open an interesting door to study the link between individual differences in behavior and detailed OFC electrophysiological encoding, by suggesting that animals that lose-switch more also have a stronger neuronal representation of past behaviorally relevant variables, and support the notion that across-subjects OFC differences modulate overall behavior, such as risk-seeking[59] and drug-seeking[60] behaviors.

## Methods

### Behavioral task.
Three Wistar rats were trained to perform an auditory time-interval categorization task. Trials were self-initiated by the animals by nose poking, which elicited a pure tone of 50 ms duration after a random delay drawn from a uniform distribution with values 50, 100, 150, 200, 250 and 300 ms. A second tone, identical in duration and frequency to the first one, was presented after a time interval, called ITI. The task is to categorize the ITI, as short ($S = s$) or long ($S = l$). ITIs are drawn randomly (see below for incorrect trials) from a uniform discrete distribution with values 50, 100, 150 or 200 ms for short intervals ($S = s$) and 350, 400, 450 or 500 ms for long intervals ($S = l$). Reward is provided in trials in which the animal sampled the full stimulus and poked to the left (right) socket, when the stimulus was short (resp. long). False alarms (poking in the opposite side) or early withdrawals (withdrawal before stimulus termination) were punished with a 3-s time out and a white noise (WAV-file, 0.5 s, 80-dB sound pressure level). After an incorrect trial, the ITI of the previous trial was repeated. This experimental design created correlations across trials based on the behavior of the animal. The mean fraction of false alarms was 0.08, 0.11 and 0.15 and the mean fraction of early withdrawals was 0.37, 0.31 and 0.14 for rat 1–3, respectively. All trials during task performance were self-initiated. The animals went through two additional passive stages before and after the decision-making stage described above. During the passive stages rats were presented with the same set of stimuli as in the decision-making stage while they could freely move around the environment. Rewards were not provided at any time during the passive stages. Passive stage A occurred before the decision-making stage and it lasted a fixed set of stimulus presentations (rat 1: 400 trials; rat 2: 600 trials and rat 3: 600 trials). Passive stage B occurred after the decision-making stage, and it lasted the same number of stimulus presentations as in passive stage A. The experiment was approved by the animal Ethics Committee of the University of Barcelona. Rats were cared for and treated in accordance with the Spanish regulatory laws (BOE 256;

25-10-1990), which comply with the European Union guidelines on protection of vertebrates used for experimentation (EUVD 86/609/EEC).

### Psychometric curve analysis.
Each rat's psychometric curve was defined as the fraction of long choices over all completed trials (correct trials and false alarms), as a function of the ITI after merging all the sessions for that animal. The all-rats psychometric curve was computed by merging all sessions from all rats. We compared the percentage of correct answers (performance) when trials were easy (ITI = 50, 100, 450 and 500 ms; far from category boundary) against the percentage of correct answers when trials were difficult (ITI = 150, 200, 350, 400 ms; close to category boundary). Significance testing of the difference of animals' performance between easy and difficult trials was based on the non-parametric bootstrap, as follows. We randomly selected with replacement $k$ trials (where $k$ is the total number of trials after merging all sessions for a particular animal or all sessions from all animals for the all-rats case) from the set of trials and assessed each rat and all-rats performances on easy and on difficult trials. We repeated this procedure 10,000 times and compared the difference of the resulting two distributions to a reference value, in this particular case zero. We defined the probability that performance on easy trials was equal to performance on difficult trials by the fraction of samples that fell above zero. The reported one-tailed $P$ values were equal to that fraction.

Psychometric curves from trials after correct (error) responses were computed by considering only those trials that followed a correct (incorrect) response. For each rat and all-rats we compared the psychometric curve after correct trials with the psychometric curve after incorrect trials. Each curve was fitted with the following function[61]:

$$P_l(\text{ITI} \mid \mu, \sigma, \gamma, \lambda) = \gamma + (1 - \gamma - \lambda)\left(\frac{1}{\sigma\sqrt{2\pi}} \int_{-\infty}^{\text{ITI}} e^{-\frac{1}{2\sigma^2}(x - \mu)^2} dx\right) \quad (1)$$

where $P_l(\text{ITI})$ is the probability of long choice as a function of the time difference between tones. The fitted parameters $\gamma$, $1 - \lambda$, correspond to the lapse rates for short ITI and long ITI respectively, whereas the parameters $\mu$ and $\sigma$ correspond to the centre and the inverse slope of the sigmoid function, respectively. We included lapse rates to avoid biased slope and centre parameter estimates[61]. The parameter estimates corresponded to the maximum likelihood solution of a binomial process with an expected value as a function of ITI defined by equation (1). We compared the steepness of the psychometric curve after correct and incorrect responses by means of the difference in inverse slope parameters $\sigma$ for the two conditions divided by the slope after correct trials (percentage change). Statistical significance was assessed by a non-parametric one-tailed bootstrap (10,000 repetitions), where we assigned uncertainty intervals to the estimated parameters and compared their difference to the reference value zero, as above. To test for significance of performance increase of the psychometric curves computed after incorrect and correct trials we used non-parametric one-tailed bootstrap as described above. The same test was used to test significance for the win-stay and lose-switch probabilities, as well as for testing if they differed.

### Neural data.
Recordings were obtained from three Wistar rats that were chronically implanted with tetrodes in their lateral orbital frontal cortex (lOFC) (Fig. 2a). We used the pre-stimulus (or trial-initiation), stimulus offset and choice periods for neuronal data analysis. The trial-initiation period starts with the rat nose-poking into the central socket and lasts for 150 ms. The stimulus offset period starts 100 ms before the second tone onset and it lasts until tone offset (150 ms in total). The choice period corresponds to a 150 ms time window that starts with nose-poking into one of the two lateral sockets.

A total of 137 single units were recorded from three rats (53, 62 and 22 from rats 1–3, respectively). On average 2.9 ± 1.6 neurons (max 8) across all rats and sessions were recorded simultaneously. We excluded all neurons firing at <1 Hz from further analysis, because their low firing rate precluded any reliable statistical analysis. All results remained qualitatively similar when including these cells. For the pre-stimulus, stimulus offset and choice periods, 76 (rat 1: 32; rat 2: 30; rat 3: 14), 87 (rat 1: 35; rat 2: 33; rat 3: 19) and 78 (rat 1: 34; rat 2: 30; rat 3: 14) single-units fulfilled the criterion, respectively (firing above 1 Hz). After filtering out low-activity units, the mean number of simultaneously recorded neurons across all rats and all sessions was 2.0 ± 1.0. Figures 2 and 3 were generated using a 100 ms causal rectangular window, sliding in steps of 50 ms. The total mean number of trials across sessions was 684, with an average number of 538 correct and 145 error trials. This led to a median of 9,000 spikes per neuron, before neuron exclusion, and a high-signal to noise ratio quality for hypothesis testing (see main text). Further details about the recordings and the experimental setup are provided in Supplementary Methods.

### ROC analysis.
For each neuron we computed the area under the curve (AUC) for a particular task variable as the probability of sampling a larger spike rate $r$ from $P(r|z = 1)$ than from $P(r|z = -1)$, where $z$ refers to any of the binary task variables[62,63]. For AUC values below one half we reversed the populations, to ensure AUCs of at least one half.

**Generalized linear model for neuronal activity.** For the GLM analysis, for each neuron we fitted the spike count in one of the three periods defined previously by:

$$n_j \sim \text{Poisson}\left(f^{-1}\left(\sum_{i=0}^{k}\omega_i x_i\right)\right) \qquad (2)$$

where the link function $f(\cdot)$ was taken to be the natural logarithm. The argument of the link function is a weighted sum over an exhaustive family of $k$ binary regressors:

$$
\begin{aligned}
\sum_{i=0}^{k}\omega_i x_i = {} & \omega_0 + \omega_1 R_{-3} + \omega_2 D_{-3} + \omega_3 C_{-3} + \omega_4 X_{-3} + \\
& + \omega_5 R_{-2} + \omega_6 D_{-2} + \omega_7 C_{-2} + \omega_8 X_{-2} + \\
& + \omega_9 R_{-1} + \omega_{10} D_{-1} + \omega_{11} C_{-1} + \omega_{12} X_{-1} + \\
& + \omega_{13} R_0 + \omega_{14} EV_0 + \omega_{15} C_0 + \omega_{16} S_0
\end{aligned} \qquad (3)
$$

Here $R_{-n}$ is the reward given to the rat $n$ trials back in time, that is, the correctness of the response ($+1$ correct, rewarded, $-1$ incorrect, non-rewarded); $D_{-n}$ is the trial difficulty defined on the basis of the distance between the presented ITI and the category boundary (50, 100, 450 and 500 ms, easy trial, $D_{-n} = +1$; 150, 200, 350 and 400 ms, difficult trial, $D_{-n} = -1$); $C_{-n}$ is rat's choice ($+1$ short choice, $-1$ long choice) and $X_{-n}$ ($n$-back second-order prior) is the interaction term between reward and choice, $X_{-n} = R_{-n} \times C_{-n}$. Thus, the variable $X_{-n}$ is also binary and it takes the value $X_{-n} = 1$, when $R_{-n}$ was correct (incorrect) and $C_{-n}$ was short (long) and the value $X_{-n} = -1$ when $R_{-n}$ was incorrect (correct) and $C_{-n}$ was short (long). For the current trial ($n = 0$), we renamed difficulty $D_0$ by $EV_0$, and refered to it as expected value, because it is of more conventional use. As $S_{-n}$ and $X_{-n}$ are the same variable, we excluded in equation (3) the former for past trials and the latter for the current trial.

The GLM fit was applied to different subsets of the data: (i) including all trials (Fig. 4) or (ii) including only trials after a correct response (Supplementary Fig. 4) and also to the datasets corresponding to the two passive stages, where the animals were presented the same set of stimuli in a passive manner (Supplementary Fig. 6). In analysis (i), the GLM included all regressors as specified in equation (3). For each regressor and neuron, statistical significance was assessed using a permutation test that sampled the null hypothesis. We shuffled each neuron's spike count across trials and fitted the model on each of 10,000 random shuffles. We defined the probability that a particular regressor was not modulating neuron's spike count by the fraction of samples that fell above or below the real regressor value for $\omega_i > 0$ or $\omega_i < 0$, respectively. Two-tailed $P$ values for each regressor and neuron were twice that fraction. The reported fraction of neurons (Fig. 4; Supplementary Figs 4 and 8) was the number of neurons that had the firing rate significantly modulated by each task-variable over the total number of neurons used in the analysis. We preferred employing a permutation to test for significance in the regressors against more traditional methods that assume that the residuals are Gaussian[20,47,51], because the residuals that we observed in our data were strongly non-Gaussian. Furthermore, permutation tests are in general more conservative (lower probability of type I errors). Finally, permutation tests sample the null hypothesis while taking into account correlations in the regressors. Note that it is not necessary to apply Bonferroni correction in our case as we always included all variables of interest in the GLM simultaneously rather than running individual tests for each variable separately.

In analysis (ii) only regressors from the previous and the current trials were included, except for $R_{-1}$ which, by construction, was constant for this particular set of trials. Regressors from previous trials were not included to avoid overfitting due to the reduced set of trials for this analysis. After correct trials, regressors $C_{-1}$ and $X_{-1}$ were equivalent and the pair was treated as a single-variable. In Supplementary Fig. 4 fractions of neurons encoding $C_{-1}$ and $X_{-1}$ were reported separately only to allow a better comparison with Fig. 4. Significance of each regressors was tested using a permutation test. We also fitted the GLM using only trials after an incorrect response. The procedure was identical to (ii) but in this case, because of the experimental protocol, $-C_{-1}$ and $X_{-1}$ and $S_0$ were identical and $EV_0$ and $D_{-1}$ were identical as well.

For the passive stages all trials were used. The set of regressors in this particular case comprised current stimulus $S_0$, current expected value or difficulty $EV_0$, second-order prior $X_{-1}$ (from $-1$ to $-3$ trials in back) and previous difficulty $D_{-1}$ (from $-1$ to $-3$ trials in back as well). It is important to note that because in the passive stage rewards are not delivered, the second-order prior variable $X_{-1}$ is undefined. However, in the decision-making stage the second-order prior variable is equivalent to the previous stimulus for all trials, that is, $X_{-1} = S_{-1}$. Thus, we take $S_{-1}$ in the passive stages as the analogous to the state-space in the decision-making task. The reported fraction of neurons (Supplementary Fig. 6) was the number of neurons that had the firing rate significantly modulated by each task-variable over the total number of neurons used in the analysis. Significance for each regressor was calculated as described above.

For each regressor a binomial test was used to assess if the fraction of neurons that had their firing rates modulated by that particular regressor was significantly greater than chance[20,51] (5%; one-tailed). Statistical significance for the difference in fractions between two conditions was tested by a non-parametric difference binomial test that sampled the null hypothesis as follows. Independent samples from two identical binomial distributions were drawn 10,000 times and the null hypothesis was built as the difference of these binomial processes. The expected values of the two identical binomial processes were the weighted mean of the two fractions to be compared. We defined the probability that the two fractions were instances of the same underlying binomial process by the proportion of samples that fell above the observed fraction difference. The reported one-tailed $P$ values corresponded to that proportion. One-tailed $P$ values were used instead of two-tailed $P$ values because the study's hypothesis was to test whether previous trial regressors (such as previous choice $C_{-1}$ or previous second-order prior $X_{-1}$) were decreasing over the course of the trial, and whether upcoming choice $C_0$ was increasing as rats went through trial's stages. For the case of upcoming stimulus $S_0$ and upcoming expected value $EV_0$ our hypothesis was that they had to peak during the stimulus presentation period.

It is important to note that it is not possible to directly compare the fractions of neurons with significant regressors after correct, incorrect or all trials, because of the large difference on the correlation structure among regressors across conditions. First, several task variables that are different on after-correct trials become the same variable for after-incorrect trials, and vice versa. For instance, $X_{-1}$ and $C_{-1}$ are the same variable after correct trials, while after incorrect trials $X_{-1}$, $-C_{-1}$ and $S_0$ are all three the same variable, and $EV_0$ and $D_{-1}$ are again the same. In addition, as depicted in Fig. 1d, rats after an incorrect response tend to switch choice more often than repeat the same choice after a correct response. Therefore, the regressor $C_{-1}$ is more strongly correlated with $C_0$ after an incorrect response than after a correct response. The differential increase of correlations between regressors, when conditioned after correct or incorrect trials and the resulting differential biases obtained from fitting a model precluded a direct comparison of the reported fractions of significant neurons across conditions.

**Correlation of regression weights.** We tested the stability of the neuronal representations over time by correlating the fitted values of weights in the GLM across different time periods. Correlations among weights could simply arise because of different responsiveness of the neurons, such that for instance when a neuron that is more responsive in the pre-stimulus period might also be more responsive in the offset stimulus period. To avoid creating correlations due to differences in overall firing rate across neurons in the population, we first normalized each firing rate by subtracting and dividing it by its mean and s.d. respectively ($z$-score) for a particular time window. This normalization can result in negative normalized rates, violating the assumptions of the previously used GLM model since a natural logarithmic function was used (equation 2). To overcome this problem, we instead fitted the data by linear regression (see previous section). Supplementary Fig 8 shows that using linear regression instead of a GLM (Fig. 4) does not qualitatively change the results. Subsequent analysis for correlated weights was performed on the linear regression coefficients, using the same set of regressors, equation 3, as for the GLM.

Stability of the neuronal representation for each variable (for example, the upcoming choice $C_0$) across the trial was assessed by using the correlation coefficient (Pearson correlation) between two vectors, each with the $i$th entry being the regression coefficient for that variable (for example, upcoming choice $C_0$) of neuron $i$, computed at two different periods, namely pre-stimulus and stimulus offset periods (Fig. 5a) or stimulus offset and choice periods (Fig. 5b). Statistical significance of the correlation coefficient was assessed by a permutation test that sampled the null hypothesis. For each regressor (for example, upcoming choice $C_0$) the null-hypothesis distribution was built from the set of correlation coefficients obtained after shuffling the relationship between each neuron's $z$-scored firing rate and the regressor, and computing their respective Pearson correlation coefficient as before. This process was repeated 10,000 times. We defined the probability that a particular regressor was not stable across time by the fraction of samples that fell above the real correlation coefficient value (if $\rho > 0$) or below the real correlation coefficient value (if $\rho < 0$). The reported two-tailed $P$ values for each regressor were twice that fraction.

We tested whether the second-order prior and upcoming choice at trial initiation are encoded by the same neurons. Unfortunately, we cannot use the same approach as just described, as computing the vectors of the regressors across neurons for both $X_{-1}$ the $C_0$, and then computing the correlation coefficient between then will lead to biases due to using two regressors from the same model in the same dataset[51]. We avoided this problem by instead computing regression weights for each variable while fixing the value of the other variable, as follows. We first restricted our analysis to trials that followed a correct response and focused on the pre-stimulus period, where only information about two variables is found, $C_0$ and $X_{-1}$ (see Supplementary Figs 4 and 8b shows how the linear regression model gives qualitatively similar results as the GLM model when focusing on trials that followed correct responses). The weights for $C_0$ were therefore computed by fitting the model on the subset of trials where the variable $X_{-1}$ was constant ($C_{-1} = X_{-1}$ for this particular set of trials; see previous section). This conditioning procedure ensured that the estimated weight for $C_0$ was not affected by its intrinsic correlation with $X_{-1}$. Because $X_{-1}$ is a binary variable, the reported weight for $C_0$ was the mean between the weight estimated for set of trials where $X_{-1} = 1$ and where $X_{-1} = -1$. The same procedure was applied for the weight associated to $X_{-1}$, where again the final weight for this variable was the mean between the weight fitted on the subset of trials where $C_0 = 1$ and $C_0 = -1$. The reported correlation coefficient was computed from two vectors, one composed of the mean

weight for $C_0$ (mean across conditionings $X_{-1} = 1$ and $X_{-1} = -1$) of each neuron $i$, and the other composed of the mean weight for $X_{-1}$ (mean across conditionings $C_0 = 1$ and $C_0 = -1$) of each neuron $i$.

Statistical significance of the correlation coefficient was again assessed by a permutation test that sampled the null hypothesis. The null-hypothesis distribution was built from the set of correlation coefficients obtained after shuffling the relationship between each neuron's $z$-scored firing rate and the regressors, and yielded one correlation coefficient sample by following the same computations as described in the previous paragraph. This process was repeated 10,000 times. We defined the probability that neurons encoding $C_0$ do not tend to encode $X_{-1}$ by the fraction of samples that fell above the real correlation coefficient value (if $\rho > 0$) or below the real correlation coefficient value (if $\rho < 0$), where $\rho$ is the correlation coefficient. The reported two-tailed $P$ values for each regressor were twice that fraction.

**Population decoding.** Small populations (two or three neurons) of simultaneously recorded single-neurons were used to classify a set of trials as belonging to either class 1 or class 2 (for example, class 1 and class 2 can correspond to short and long choices for the variable $C_0$, or to correct and incorrect responses for the variable $R_{-1}$). Classification is based on a decision variable DV: when $DV > 0$ the trial is classified as class 1, and when $DV < 0$ the trial is classified as belonging to class 2. The decision variable DV is a weighted sum of the population activity $DV = \sum_{i=1}^{N} \omega_i r_i + \omega_0$, where $\omega_i$ and $r_i$ are each neuron's contribution to the decision variable and spike rate respectively, $\omega_0$ is the offset term, and $N$ is the total number of neurons used in the classifier. Logistic regression assumes that the probability of class 1 to be the correct class given the activity pattern of the population is given by $p(\text{class1}|\{r_i\}) = \sigma(\sum_{i=1}^{N} \omega_i r_i + \omega_0)$, where $\sigma(\cdot)$ is the logistic function. The model was trained and tested using five-fold cross validation.

For most sessions, the number of trials belonging to class 1 did not match the number of trials belonging to class 2, in other words, conditions were unbalanced. We addressed this problem by subsampling[64,65], which consists in balancing the number of trials for the two classes by randomly excluding trials from the most populated class. A large imbalance can be problematic when comparing classifier's performance among data sets: if class 1 and class 2 are unbalanced, then Decoding Performance (DP) can be larger than chance (DP = 0.5) even when there is no information in any of the regressors. Subsampling was repeated 20 times. Each time the model was trained and tested by 5-fold cross validation. The reported decoding performance (DP; fraction of correct classifications) corresponds to the mean DP over all recording sessions, subsampling and cross-validation iterations.

Statistical significance of DP was tested using a permutation test that sampled the null hypothesis. For the set of trials (the whole recording session when class 1 and class 2 were balanced and the particular subsampling iteration when class 1 and class 2 were unbalanced) we shuffled each trial's class label and estimated DP through the five-fold cross-validation method (20 repetitions for the subsamplings). This procedure was repeated 1,000 times. Each of the samples of the null hypothesis distribution was computed as the mean across recording sessions, subsampling and cross-validation for a particular shuffling iteration. We defined the probability that the neuronal ensemble had no information about that particular task variable by the fraction of samples that fell above the real DP. The reported one-tailed $P$ values were that fraction.

**Conditioned population decoding.** As many of the variables are partially correlated (for example, choice with stimulus), being able to decode one of them necessarily means that we can decode the others. To test if we can read out both of a pair of partially correlated variables independently, we performed a conditioning decoding analysis in which we tested for information of one variable while keeping the values of the other variable fixed (Fig. 6). We restricted our analysis to trials after correct responses. As shown in Supplementary Fig. 4, the GLM analysis revealed that single-neurons seemed to encode only two variables: upcoming choice $C_0$ and second order prior $X_{-1}$. We therefore decoded upcoming choice $C_0$ by fitting a classifier on the subset of trials where $X_{-1} = 1$ and $X_{-1} = -1$ independently (subsampling method and five-fold cross validation, see previous section). The reported DP when classifying upcoming choice given second-order prior was the mean between the two conditioned DP. To decode $X_{-1}$ the same procedure was applied but conditioning on each of the two possible values of $C_0$ instead. The reported DP when classifying second order prior given upcoming choice was the mean between the two conditioned DP. In this way, even though decoded quantities might be correlated, reported population information content about $C_0$ and $X_{-1}$ could not be explained simply by a correlation to other variables (Fig. 6). $P$ values were computed using a permutation test, as described in previous section.

**Information ranking.** We used decoding performance (DP) for each variable that was deemed significant by the GLM analysis as a proxy for the amount of information that the neuronal population contained about that variable (Fig. 4). DP is computed as described above. Our analysis provides the intuitive result that decoding performance increases with the number of neurons in the ensemble (Figs 6 and 7). Some previous population analysis violated this due to misusing linear classifiers[66].

**Data availability.** The datasets generated in this study and the code used for their analysis are available from the corresponding author upon reasonable request.

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

## Acknowledgements

R.N. is supported by a FI-AGAUR scholarship from the Government of Catalonia. R.M.-B. is supported by PSI2013-44811-P and FLAGERA-PCIN-2015-162-C02-02 from MINECO (Spain). M.V.S.-V. is supported by BFU2014-52467-R and SlowDyn FLAGERA-PCIN-2015-162-C02-01 from MINECO. This work was supported by CERCA Programme / Generalitat de Catalunya. We thank Julio Martinez-Trujillo for comments on the manuscript.

## Author contributions

R.N. performed the analysis and generated results and J.M.A. performed the recordings. All authors designed the study, discussed results and wrote the paper.

## Additional information

**Competing interests:** The authors declare no conflict of interest.

