## [Peer Review File · Nature Communications]

Reviewers' comments:

Reviewer #1 (Remarks to the Author):

This manuscript by Nogueira, Aboliafia, et al. reports results from a single unit study in the orbitofrontal cortex (OFC) of rats trained to perform a perceptual discrimination. One of the key features of the task is that a stimulus was repeated after an incorrect response, whereas new stimuli were chosen from a uniform distribution following correct responses. As a result, a win-neutral, lose-shift heuristic was optimal. The authors report that single units and small neural populations encoded the conjunction of previous outcome and choice, upcoming choice, and expected value/confidence. Of these variables, information about upcoming choice and the conjunction of past history and outcome was most easily extracted from population activity.

This is a solid study. The question of OFC's role in decision making is an important one. The analyses are reasonable. However, the design of the experiment, coupled with the contentious multitude of live hypotheses in current OFC research, make the interpretation of this study difficult. This is another valuable data point in what is now an intricate debate, and while it may add meaningfully to that discussion, I am not sure it helps resolve the outstanding issues.

To elaborate:

- The authors report that the conjunction of past choices and outcomes is strongly encoded in OFC. This seems consistent with past results from the Rushworth and Schoenbaum groups (I'm thinking particularly of Noonan et al. 2010 and Walton et al. 2010) about the importance of context in OFC, but I'm unclear where the novel contribution lies. This may simply be something I'm overlooking in the presentation, but clarifying this would help the present ms.

- Similar things could be said for other reported variables: expected value (Padoa-Schioppa and Assad), confidence (Kepecs, Uchida, and Mainen), stimulus type (Watson and Platt). I gather from the text that the authors have a somewhat different take on the role that these variables might play in the decision process than previous authors, but it is hard for me to know how to rule out any of those previous stories, which in many cases could be told about these same data. Current theories of OFC are so flexible that it is not hard to imagine using these new data to add limited support for any one of them.

- There are two other phenomena that are difficult to compensate for but should be acknowledged by the authors:

- It is quite possible that the authors find preferential encoding of X and C0 due to the fact that these were the most relevant variables for task performance. Compare, for instance, the encoding of reward in Padoa-Schioppa and Assad to that in Watson and Platt, where reward was less salient. Some aspects of cortical responsiveness are likely to respond to the overtraining animals undergo, and neural codes, especially prefrontal ones, are subject to some task demands. It is very difficult to rule such things out, and this is not a particular limitation of the present work, but should be taken into account.

- It is likewise possible that the presence of information about upcoming choice in OFC is a

result of network effects and reentrant activity, whether or not OFC is involved in action selection. Again, it is difficult to disambiguate this using purely correlational data, but it should be acknowledged that OFC could carry upcoming choice-related signals without itself being involved in action selection.

Reviewer #2 (Remarks to the Author):

In this work Nogueira et al. look at the representation of decisions-related variables in OFC. They use a novel duration estimation task with an interesting choice-history-dependent structure in which rats are required to repeat incorrect trials. This history-dependence allows rats to perform better on the task if they take into account their previous choices and, behaviorally they appear to adopt a lose-shift policy which is optimal for this task (and to a lesser extent a suboptimal win-stay strategy too). In line with their behavior, neurons in OFC appear to code all the relevant variables required to implement this lose-shift policy (previous choice, previous reward and their interaction) throughout the task. Moreover, they also find evidence that OFC codes the upcoming choice, a variable that has not often been seen in OFC.

I think this is a nice paper and the analyses are really nice (although I think one very important analysis might be missing). Many of the findings are consistent with the literature and existing theories of OFC. The finding that OFC codes upcoming choice is slightly surprising and I have one substantive concern about this. Apart from that my comments are relatively minor pointing to a couple of missing refs and the relatively low N.

Major concern

1. The coding of upcoming choice in OFC is intriguing. While this is not unprecedented (see Rich and Wallis ref below) I think there is a potential issue with this interpretation in this task. In particular, as the authors point out, coding the last choice would be advantageous in a task like this because when that choice was wrong the rat should just "lose-switch" to the correct answer. Thus the rat should be coding previous choice, which (when that choice was not rewarded) will be the opposite of current choice.

So my question is: Is C0 encoded equally in OFC on R-1 = +1 and on R-1 = -1 trials? The authors themselves raise a similar concern about coding S0 before the stimulus is actually presented. They resolve the S0 issue by looking at R-1 = +1 trials only and find S0 is not encoded. I wonder if they could do the same thing for C0?

Minor comments

1. Low N. The number of rats is small (3) and the number of neurons isn't enormous either (if I'm reading it right - 76 after exclusions). It would obviously be better to have more animals and neurons, but I realize this may not be an option. An alternative to running more subjects would be to work through a simple power analysis to show how many

animals/neurons you would need for a sufficiently powered replication of this. Such an analysis could easily be left to supplementary material but it would give a good sense of how much we should worry about $N = 3$.

2. There are a couple of refs that should be in here

Rich, E. L., & Wallis, J. D. (2016). Decoding subjective decisions from orbitofrontal cortex. *Nature neuroscience*.

This paper shows that ensemble activity in OFC vacillates between representations of the different options in a choice paradigm. It would be interesting to discuss your finding that choice is represented in OFC in relation to this.

SCY Chan, Y Niv* & KA Norman* (2016) – A probability distribution over latent causes in the orbitofrontal cortex – *The Journal of Neuroscience* 30(30):7817-7828

This is a relevant fMRI paper showing something like a posterior (which is the product of prior and current information) in OFC.

Typos

"including all cells did not qualitative" -> qualitatively

Page 10 – "expended value" -> expected value

Page 12 - "correlation was not reach significance" -> "did not"

Reviewer #3 (Remarks to the Author):

The argument put forth in the present manuscript is that latent information about choice history is encoded in the OFC and then used to control future decision-making. This has been an unanswered question in the field and has been more elusive to understand. They use a perceptual task where stimulus presentation of varying difficulties signaled the potential delivery of an appetitive outcome, and subjects' choice history controlled the presentation of future events. Extracellular recordings of OFC neurons were made and a hidden Markov model revealed encoding of choice, prior information, and stimulus information. The observation of latent information build-up over time in OFC states was more novel and very interesting. Data presented show a careful analysis of OFC signals and provide interesting hypothesis about OFC involvement in maintaining a task space based on recent decision-making history. This was a great analysis and task to investigate this question, and the findings are exciting. The lack of data showing a functional contribution of this coding limits enthusiasm.

-The recent report by Schuck et al., in *Neuron* reports decoding of task space in humans. Please discuss your findings in relation to this recent report.

-There is a lack of functional data suggesting that the OFC encoding and build up of choice

history and increasing observed is necessary for choice execution. Can the authors comment on what the expectations are for a functional role? For example, what could one expect in OFC activity was disrupted or inhibited prior to stimulus onset? This seems to be testable hypothesis in this modern age in rodents.

-The finding of OFC involvement in action initiation and selection based on current choice information is not novel; the analysis presented here on the representation of the immediate history is. There have been numerous published reports that OFC is involved in action selection (e.g., Gourley et al., 2013; Gremel & Costa, 2013; Rhodes & Murray, 2013; Bradfield et al., 2015; Gremel et al., 2016). Importantly, OFC involvement depended on actions where the immediate expected outcome was controlling action selection. OFC encoding of action selection prior to action initiation has been shown- and has been shown to be functionally relevant. Bradfield et al., 2015 nicely showed that OFC was functionally necessary to retrieve action consequence information and use that information to guide behavior where the action consequences were not visible but had to be inferred from recent experience. Perhaps the authors need to reframe the underlying novelty their analysis brings to the table.

-One major concern is the limited number of neurons sampled across only 3 rats. Neuron firing is assessed across only 3 rats; with Further, only a small number of neurons were sampled in each animal. That number was reduced almost in half when looking at neurons that showed significant responses, and the final number of neurons in each animal that showed changes was not reported. Further, since firing was assessed and collapsed across many trials, there is a concern that the same neurons were continuously sampled and this sampling could skew the proportional data as well as bias response data. What was done to minimize this during data collection, or would the addition of rats to the analyses strengthen the data?

Reviewers' comments:

Reviewer #1 (Remarks to the Author):

1. *“This manuscript by Nogueira, Aboliafia, et al. reports results from a single unit study in the orbitofrontal cortex (OFC) of rats trained to perform a perceptual discrimination. One of the key features of the task is that a stimulus was repeated after an incorrect response, whereas new stimuli were chosen from a uniform distribution following correct responses. As a result, a win-neutral, lose-shift heuristic was optimal. The authors report that single units and small neural populations encoded the conjunction of previous outcome and choice, upcoming choice, and expected value/confidence. Of these variables, information about upcoming choice and the conjunction of past history and outcome was most easily extracted from population activity. This is a solid study. The question of OFC’s role in decision making is an important one. The analyses are reasonable. However, the design of the experiment, coupled with the contentious multitude of live hypotheses in current OFC research, make the interpretation of this study difficult. This is another valuable data point in what is now an intricate debate, and while it may add meaningfully to that discussion, I am not sure it helps resolve the outstanding issues.”*

We would like to thank the reviewer for his/her critical view. Below we have responded to all your questions.

2. *“The authors report that the conjunction of past choices and outcomes is strongly encoded in OFC. This seems consistent with past results from the Rushworth and Schoenbaum groups (I’m thinking particularly of Noonan et al. 2010 and Walton et al. 2010) about the importance of context in OFC, but I’m unclear where the novel contribution lies. This may simply be something I’m overlooking in the presentation, but clarifying this would help the present ms.”*

We thank the reviewer for this question, and for the opportunity to clarify the novelty of our results. In summary, the novelty of our results lies in showing (electrophysiological evidence) that:

1. Choice-related neuronal signals appear even before stimulus onset (this result has not been shown previously, except in a previous monkey report by Padoa-Schioppa, which is cited in the manuscript).
2. OFC integrates ambiguous sensory information with prior information (our work develops a novel paradigm that studies the effect of immediate past information in

perceptual decision-making, characterized in detail by their effects on the psychometric curves).

3. OFC can represent prior information exclusively from the previous trial. This means that the representation of the state of OFC has ‘high temporal resolution’ (see next).

Regarding the last point, the reviewer is absolutely right that it has been previously shown that OFC tracks previous choices and outcomes (Noonan et al and Walton et al). Similarly, as pointed out by the reviewer, a large body of work by Schoenbaum also shows that OFC is important for learning stimulus-reward associations. All this body of knowledge emphasizes learning associations between choices and outcomes over a long history of observations (substantially more than one trial back in the past, and thus pertaining to intermediate or long timescale learning). However, whether OFC is flexible enough to track critical state variables in the world that can rapidly change on a trial-by-trial basis at ‘high temporal resolution’ (short timescale) has been elusive. To our knowledge, our work is the first one that suggests that OFC represents prior information mostly from the previous trial. We have fully rewritten the Discussion section to clarify these important points.

3. “Similar things could be said for other reported variables: expected value (Padoa-Schioppa and Assad), confidence (Kepecs, Uchida, and Mainen), stimulus type (Watson and Platt). I gather from the text that the authors have a somewhat different take on the role that these variables might play in the decision process than previous authors, but it is hard for me to know how to rule out any of those previous stories, which in many cases could be told about these same data. Current theories of OFC are so flexible that it is not hard to imagine using these new data to add limited support for any one of them.”

This is also a valid concern, which probably applies to most OFC research. Note, however, that our data indeed favors the notion that OFC generally encodes only variables that are relevant given the task structure (the so called ‘state-space’), an idea that has been put forward recently by Niv and collaborators. But our data also support the notion that OFC might also participate in action selection by taking advantage of the state-space encoding, as we have uncovered choice-related signals very early in the decision process. Thus, our data suggest that OFC represents the state-space (Niv’s et al idea) and adds that this representation interacts with the decision-making process in OFC. Further, our data does not directly support the simpler notions that OFC is just the site of value or confidence encoding.

We have added the Watson and Platt reference in the introduction, which was not referred to in the previous version. We have also added text in the Discussion section to clarify this important point.

4. “There are two other phenomena that are difficult to compensate for but should be acknowledged by the authors:

It is quite possible that the authors find preferential encoding of X and C0 due to the fact that these were the most relevant variables for task performance. Compare, for instance, the encoding of reward in Padoa-Schioppa and Assad to that in Watson and Platt, where reward was less salient. Some aspects of cortical responsiveness are likely to respond to the overtraining animals undergo, and neural codes, especially prefrontal ones, are subject to some task demands. It is very difficult to rule such things out, and this is not a particular limitation of the present work, but should be taken into account.”

This is indeed an extremely important comment, most typically overseen by PFC studies using over-trained animals. We believe that the flexibility of OFC is very important for testing the hypothesis that OFC neuronal code represents relevant state variables and that this structure participates in the decision-making process. This is why we choose a task in which all these elements could be studied at once (prior information from the previous trial, confidence, choice, stimulus, etc.), while most previous research has narrowed down the research task to the testing of a few variables (such as value or reward). We thus agree that our data support the notion that OFC representation is flexible, but this is indeed the question that we wanted to test. To further address the reviewer’s question, we have also added new data showing that, interestingly, OFC does not represent immediate past variables when these variables are not behaviorally relevant for the animal (see Supplementary Fig. 6).

5. *“It is likewise possible that the presence of information about upcoming choice in OFC is a result of network effects and reentrant activity, whether or not OFC is involved in action selection. Again, it is difficult to disambiguate this using purely correlational data, but it should be acknowledged that OFC could carry upcoming choice-related signals without itself being involved in action selection.”*

We agree with the reviewer’s point that the signals that we have characterized in OFC can be due to reentrant activity from other areas. This is also a valid concern for most electrophysiological research. One way of addressing this issue is by identifying the region (or regions) that first show a given signal. In our case, we have found that OFC represent choice-related signals way before stimulus onset. This result contrasts with previous failures of finding early choice-related signals in the OFC. These failures were taken as evidence that the OFC does not participate in action selection (refs 14, 20, 21 in our manuscript). Therefore, our characterization of early choice-related signals in OFC opens the door to the possibility that OFC does in fact participates in action-selection.

Reviewer #2 (Remarks to the Author):

1. *In this work Nogueira et al. look at the representation of decisions-related variables in OFC. They use a novel duration estimation task with an interesting choice-history-dependent structure in which rats are required to repeat incorrect trials. This history-dependence allows rats to perform better on the task if they take into account their previous choices and, behaviorally they appear to adopt a lose-shift policy which is optimal for this task (and to a lesser extent a suboptimal win-stay strategy too). In line with their behavior, neurons in OFC appear to code all the relevant variables required to implement this lose-shift policy (previous choice, previous reward and their interaction) throughout the task. Moreover, they also find evidence that OFC codes the upcoming choice, a variable that has not often been seen in OFC. I think this is a nice paper and the analyses are really nice (although I think one very important analysis might be missing). Many of the findings are consistent with the literature and existing theories of OFC. The finding that OFC codes upcoming choice is slightly surprising and I have one substantive concern about this. Apart from that my comments are relatively minor pointing to a couple of missing refs and the relatively low N.*

We would like to thank the reviewer for the support and for all the comments provided below.

2. *Major concern. The coding of upcoming choice in OFC is intriguing. While this is not unprecedented (see Rich and Wallis ref below). I think there is a potential issue with this interpretation in this task. In particular, as the authors point out, coding the last choice would be advantageous in a task like this because when that choice was wrong the rat should just “lose-switch” to the correct answer. Thus the rat should be coding previous choice, which (when that choice was not rewarded) will be the opposite of current choice. So my question is: Is C0 encoded equally in OFC on R-1 = +1 and on R-1 = -1 trials? The authors themselves raise a similar concern about coding S0 before the stimulus is actually presented. They resolve the S0 issue by looking at R-1 = +1 trials only and find S0 is not encoded. I wonder if they could do the same thing for C0?*

Thank you a lot for these comments. First, we would like to briefly acknowledge the importance the recent Rich and Wallis paper, which is now referred to in the Discussion section. We would like to mention, however, that there is an important difference between that paper and our work: while in the Rich and Wallis paper encoding of upcoming choice happens after stimulus onset, we find encoding of upcoming choice before stimulus onset. This is an important novelty, as it shows that choice-related signals can arise very early in OFC, suggestive of a role of OFC in combining prior with sensory information for action selection.

Regarding your predictions, this is indeed a very important point that we are totally eager to discuss with you. As a matter of fact, we have been thinking about this and related

predictions in detail from the very beginning of our analysis. As you suggest, the analysis reported in Fig. 4 of the main paper can be applied to trials following a correct response of the rat ($R-1 = +1$) as well to trials following an incorrect response ($R-1 = -1$). In both cases, we can apply a linear model to estimate the fraction of neurons encoding significantly each task variable. However, our enthusiasm for this analysis was dampened by the following two reasons:

- 1) The number of trials used to fit each model is a key factor for the results. As described in our paper our results are based on a permutation test that samples the null hypothesis after shuffling the relationship between the regressors and the neuron's activity. We consider that a neuron is significantly encoding a given regressor if the original fitted parameter lies within the 2.5 upper (or lower) percentile of the null hypothesis distribution. The more trials a dataset has, the narrower will be the null hypothesis distribution and therefore the percentage of type II errors (misses) will be reduced. The smaller the number of trials the larger the number of type II errors. Therefore, the difference in the number of trials is a key component to control for when comparing fractions of neurons of two different datasets.
- 2) The other key factor is the correlation structure of the regressors (independent variables) used to fit the neuronal activity (dependent variable). Correlated regressors make parameter estimates noisier and therefore the null hypothesis distribution also has a larger dispersion when correlations between the regressors are large. As above, the larger the correlation structure the more prone will be our model to commit type II errors and therefore underestimating the fraction of neurons encoding for a given regressor.

Now we can apply these two considerations to our data. First, in our dataset, the set of correct trials ($R-1 = +1$) is much larger than the set of error trials ($R-1 = -1$), as the rats perform much better than chance (Fig. 1 in our manuscript). Second, in our dataset, the correlation structure of the regressors after an incorrect response is larger than after a correct response because rats follow predominantly a lose-switch over a win-stay strategy (Fig. 1 in our manuscript) and therefore after an incorrect response $C0$ and $C-1$ are more strongly correlated ($\rho < 0$) than after a correct response. Therefore, making comparisons between the fraction of neurons after correct and after incorrect can be misleading.

Controlling for point 1 above can be done by randomly selecting trials from the most populated class (after correct response) so that it matches the number of trials for the least populated class (after incorrect), at the cost of increasing the probability of false negatives (type II error). The model can be fitted and the fraction of neurons for each regressor can be calculated for a given iteration. The left panel in Figure 1 (see below) shows the same analysis than performed in Supplementary Figure 4 in the paper after controlling for the difference in the number of trials (the reported fraction corresponds to the mean across iterations). When compared to Supplementary Figure 4, the fraction of significant neurons shows a drop for most of the regressors, confirming that a lower number of trials increases the probability of type II errors (underestimating the real fraction of neurons).

Figure 1. Fraction of neurons encoding the different regressors after correct (left panel) and incorrect (right panel) responses. The number of trials for both datasets has been equalized through a subsampling procedure (see text).

Even though the difference in the number of trials can be controlled (point 1 above), unfortunately controlling for the structure of correlations of the set of regressors cannot be done (point 2 above). The correlation structure is something defined by rats' behavior and it cannot be modified offline. Therefore, the larger correlation structure among regressors after an incorrect response makes the type II errors much more frequent than after correct responses, making any quantitative comparison of fractions between the two conditions misleading.

To make a more concrete example and visualize better this issue, we ran a simulation where we created surrogate neuronal responses of a population of neurons that depended on the regressors from the data. Importantly, we used the same set of regressors than in the real data, thus keeping intact their correlation structure. For each neuron in the population and each trial, we generated a spike count based on the weighted sum of all the trial's regressors that could be modulating neuron's firing rate. In each trial the spike count of a neuron was sampled from a Gaussian distribution with mean being the weighted sum of regressors. The weights for the regressors (for a particular neuron) were sampled from a Gaussian distribution. After generating the matrix of regressors (number of neurons \times number of regressors), a fraction of them were set manually to zero (each column of the matrix was set a given fraction of its values to zero). In this way, we could control the ground truth fraction of neurons encoding for any given regressor. Once the surrogate data for each neuron was generated we fitted our linear model and reported the obtained fraction of neurons encoding for each regressor (as in Fig 4 of the main paper) with the main difference that now we know the ground truth for each variable.

Figure 2 shows that, even though the ground truth fraction of neurons encoding for C_0 , X_{-1} and C_{-1} after an incorrect response is slightly larger than after a correct response (compare pale bars, ground truth), the estimated fractions of neurons encoding for the regressors is larger after correct than after incorrect responses (compare solid bars, estimates). As stated before, this is because after an incorrect response, the probability that the inferred model

leads to a type II error -thus underestimating the fraction of neurons- is larger than after correct responses. It is important to note that in this case we also controlled for the number of trials as we explained above. Therefore, even though after an incorrect response there was a larger fraction of neurons encoding for C_0 , C_{-1} and X_{-1} , the results of a linear model analysis showed results pointing in the opposite direction.

Figure 2. Fraction of neurons encoding the different regressors after correct (left panel) and incorrect (right panel) responses using surrogate neuronal activity. The number of trials for both datasets has been equalized through a subsampling procedure as in Figure 1 above. Even though the ground truth after incorrect trials is larger (pale bars), due to an increase of type II errors, a linear model analysis estimates a larger fraction of neurons after a correct response (solid bars).

For all the reasons stated above, we think that it is misleading to make inferences based on quantitatively comparing fractions for neurons across very different conditions.

As a final remark, we would like to mention that we performed a similar analysis comparing fractions of neurons for upcoming easy trial vs upcoming difficult trial. If neurons in OFC do integrate immediate prior with sensory information in a perceptual decision-making task, on easy trials the amount of information about immediate prior should be smaller than on difficult trials. This is because on difficult trials sensory information is less relevant and therefore rats should rely more on past trial information to make a correct choice.

Figure 3 suggests that indeed there is a larger number of cells encoding immediate prior during difficult than during easy trials. Even though this result could naively be taken as evidence in favor of our hypothesis, it was discarded because of the same reason we discarded the results in Figure 1 and Figure 2 above. Finally, we would like to note that even though this issue was already mentioned in the Supplementary Information in the previous version of the manuscript, we rewrote it in a clearer way.

Figure 3. Fraction of neurons encoding for the different regressors for easy (left panel) and difficult (right panel) trials. The number of trials for both datasets has been equalized through a subsampling procedure, as in Figure 1.

Minor comments

3. *Low N.* The number of rats is small (3) and the number of neurons isn't enormous either (if I'm reading it right – 76 after exclusions). It would obviously be better to have more animals and neurons, but I realize this may not be an option. An alternative to running more subjects would be to work through a simple power analysis to show how many animals/neurons you would need for a sufficiently powered replication of this. Such an analysis could easily be left to supplementary material but it would give a good sense of how much we should worry about $N = 3$.

Thank you a lot for this suggestion, which we follow in full below. The power analysis both at the level of neurons and animals has been included in Supplementary Information, and strongly supports our claims in the manuscript.

Power Analysis. To address whether the sample size used in this study was sufficiently large, we performed a power analysis. The power (π) in hypothesis testing is defined as the probability of correctly rejecting the null hypothesis (when it is false). It is related to the β , the probability of a type II error, through the equality $\pi = 1 - \beta$.

We first calculated the statistical power of the fraction of neurons that had their firing rate significantly modulated by each one of the reported regressors. Because significance for each neuron and regressor was calculated using a Binomial test (one-tailed), the power analysis was calculated using a Binomial distribution as well. First, we calculated the threshold on the fraction of neurons that would make us reject the null hypothesis when it is true. This value was the smallest fraction of neurons that fulfilled $p \leq 0.05$ for a one-tailed Binomial test. As this quantity only depends on the total number of neurons used in the Binomial test, it is the same across all regressors. We calculated the statistical power of the fraction of neurons encoding for a particular regressor as the probability of sampling a largest or equal value than

the threshold from a binomial distribution, with probability of success defined as the actual fraction of neurons reported in the study. For a fixed probability of type I error ($\alpha = 0.05$) the statistical power increases with the strength of the result itself (reported fraction of neurons) and the number of samples used in the Binomial test (number of neurons). In other words, the probability of making a type II error when inferring the significance of that result decreases if these quantities increase.

The same procedure was applied to the statistical power analysis associated with the number of rats. Every rat that showed results in favor of the hypothesis was considered a success, and otherwise it was a failure. Because all rats in the study belonged to the success class (our main results are consistent across rats) we calculated the power if an additional rat without significant results (belonging to failure class) was added to the study.

We considered a result had sufficient statistical power when it exceeded the standard threshold (Baussell Barker, R. and Yu-Fang Li, 2002) $\pi = 0.80$. For the number of single units ($N=76$), the statistical power of the fraction of neurons encoding C_0 before stimulus presentation was $\pi=0.9996$, and $\pi=0.9947$ for X_{-1} . Thus, the two central results of our study are statistically rather solid.

Regarding the number of rats, we found significant results consistently across all rats. If we included a fourth rat with negative results the power of our conclusions would be $\pi = 0.95$. Thus, despite the small sample size, the strength of the results supported sufficient statistical power.

To directly address the reviewer's question about what would be the smallest number of neurons and rats for a sufficiently powered replication of our results, we proceeded in the following way: we assume that the fraction of cells we have reported is the ground truth, and calculated the necessary number of total neurons such that $\pi = 0.80$.

Given that the reported fraction of neurons for C_0 before stimulus onset was 0.25, if $N = 21$ we would find $\pi = 0.81$. For X_{-1} , the fraction of neurons before stimulus onset was 0.21, so with $N = 25$, we would reach a sufficient power. For C_{-1} the fraction was 0.30, so $N = 18$ would also yield enough power. Therefore, a sufficiently powered replication of our main findings would only require at least $N = 25$ neurons, well below the 76 units considered.

Regarding the number of rats, if we had four rats and assuming that the fourth rat did not produced results in the direction of our hypothesis, then the statistical power of our analysis would still be $\pi = 0.95$, as stated above. However, if only 2 out of 3 rats did produce hypothesis-aligned results, the power would not be sufficient, as $\pi = 0.75$. Therefore, three rats would be needed for a sufficiently powered replication of our results.

4. *There are a couple of refs that should be in here. Rich, E. L., & Wallis, J. D. (2016). Decoding subjective decisions from orbitofrontal cortex. Nature neuroscience. This paper shows that ensemble activity in OFC vacillates between representations of the different*

options in a choice paradigm. It would be interesting to discuss your finding that choice is represented in OFC in relation to this.

Thanks a lot for pointing to us this recent and very relevant paper. In this paper Rich and Wallis find that the activity of neuronal population represents over time the values of one of the two offered options. This intriguing result is consistent with OFC playing a role in internally deliberating between the two options, and in this regards it is consistent with our result that there are strong choice-related signals in OFC. As we said also above, we now comment on this paper in the Discussion section.

5. SCY Chan, Y Niv & KA Norman* (2016) – A probability distribution over latent causes in the orbitofrontal cortex – The Journal of Neuroscience 30(30):7817-7828. This is a relevant fMRI paper showing something like a posterior (which is the product of prior and current information) in OFC.*

Thanks again. This is another interesting paper, related to our result that OFC integrates sensory with prior information, and thus consistent with Chan et al results that OFC encoded a posterior probability distribution. We have commented on this related paper in the Discussion section

6. Typos: “including all cells did not qualitative” -> qualitatively

Page 10 – “expended value” -> expected value

Page 12 - “correlation was not reach significance” -> “did not”

Thanks a lot!

Reviewer #3 (Remarks to the Author):

1. *The argument put forward in the present manuscript is that latent information about choice history is encoded in the OFC and then used to control future decision-making. This has been an unanswered question in the field and has been more elusive to understand. They use a perceptual task where stimulus presentation of varying difficulties signaled the potential delivery of an appetitive outcome, and subjects' choice history controlled the presentation of future events. Extracellular recordings of OFC neurons were made and a hidden Markov model revealed encoding of choice, prior information, and stimulus information. The observation of latent information build-up over time in OFC states was more novel and very interesting. Data presented show a careful analysis of OFC signals and provide interesting hypothesis about OFC involvement in maintaining a task space based on recent decision-making history. This was a great analysis and task to investigate this question, and the findings are exciting. The lack of data showing a functional contribution of this coding limits enthusiasm.*

We would like to thank the reviewer for the detailed list of comments and suggestions. We have followed point by point your suggestions, and hope that you find them appropriate.

2. *The recent report by Schuck et al., in Neuron reports decoding of task space in humans. Please discuss your findings in relation to this recent report.*

Thanks for pointing us to this important paper, which is directly related to the novel hypothesis by Niv and collaborators that OFC represents a cognitive map of state-space (we have quoted the work by Niv and collaborators acknowledging this point in the previous version of the manuscript, but we have this point clearer in the new version of the manuscript). The paper by Schuck et al adds additional support to that view, and we have added it as a reference in the Discussion section. This work is generally consistent with our result that OFC encodes state-space in the form of the second-order prior variable $X-1$. The novelty of our work is to show that the representation of state-space is of high temporal resolution, as in our task task's state-space is defined by the immediately preceding trial variables. We have further clarified this important point further throughout the manuscript.

3. *There is a lack of functional data suggesting that the OFC encoding and build up of choice history and increasing observed is necessary for choice execution. Can the authors comment on what the expectations are for a functional role? For example, what could one expect in OFC activity was disrupted or inhibited prior to stimulus onset? This seems to be testable hypothesis in this modern age in rodents.*

We have added new data to further support our finding that OFC represents state variables for the integration of sensory with prior information from the previous trial. We think that this new data provide further support for a role of OFC on information integration.

In summary, the rats in our new study went through three main stages. The first stage, “passive stage A”, consisted in the rats being presented with the same stimulation protocol as the one described in the Methods section, but rats could freely move around the environment and they were not required to make any decision. During the second stage, “engaged” or “decision-making stage”, rats were presented with an outcome-coupled Markov task as described in the Methods. All the results described in our manuscript belong to this stage. The third stage, “passive stage B”, was identical to “passive stage A”. As we show next, these passive stages, not analyzed before, provide very important clues about the functional role of OFC in integrating prior with sensory information.

Figure 1 below shows the fraction of neurons encoding several relevant variables during the passive stages A and B. For the analysis, we applied a GLM model as described in the Methods section (Eq. 3 and 4). Regressors used for the linear fit comprised current stimulus S_0 , current expected value or difficulty EV_0 , second-order prior (from -1 to -3 trials in back) and previous difficulty (from -1 to -3 trials in back as well). It is important to note that second-order prior is equivalent to previous stimulus for this task, and thus both names will be indistinctly used from now on. We report averaged fractions across the two passive stages. The total number of neurons used for this analysis was the same as for Figure 4 in the paper. Trial-imitation and stimulus-offset periods were defined as in the Methods section of the paper. The post-stimulus period consisted on a 150 ms time window after stimulus presentation.

Figure 1. Fractions of neurons encoding current stimulus S_0 , expected value EV_0 , second-order prior X_{-1} and previous difficulty D_{-1} at trial initiation (pre-stimulus), stimulus offset and post-stimulus periods. Shaded rectangle corresponds to non-significant fraction of neurons ($p > 0.05$).

The crucial results are as follows: during the passive states, neurons exclusively encoded the current stimulus. This means that in this case, variables such as the second-order prior X_{-1}

(e.g., previous stimulus) were ignored and not represented in OFC (at least at the temporal and spatial resolution of our recordings). These results contrast with those in Fig. 4 in the main text, where it is shown that second-order prior is one of the most strongly encoded variables in OFC. Overall, these results suggest that OFC is a structure that represents state variables when they are relevant to the task at hand, and they are filtered out when they are no longer relevant. We have added these new data in the Results section (Supplementary Fig. 6)

Finally, we are happy to comment on the potential impact of inactivating or disrupting OFC activity during our task. Our data suggest that OFC is important for bringing together the most recent relevant past with the current stimulus. Therefore, inactivation of OFC could result in an impairment of combining the information from the previous trial with the stimulus information in the current trial. This impairment could be partial or total depending on the involvement of OFC in the action selection process. In particular, we would expect a reduction of lose-switch (see Fig. 1 in the paper).

However, it is also possible that in over-trained animals information is redundant and represented in multiple brain areas, in which case there might not be an impairment following OFC inactivation. However, we fully agree with the reviewer that confirmation or refutation of these predictions will add further information about the functional role of OFC in perceptual decision-making.

4. The finding of OFC involvement in action initiation and selection based on current choice information is not novel; the analysis presented here on the representation of the immediate history is. There have been numerous published reports that OFC is involved in action selection (e.g., Gourley et al., 2013; Gremel & Costa, 2013; Rhodes & Murray, 2013; Bradfield et al., 2015; Gremel et al., 2016). Importantly, OFC involvement depended on actions where the immediate expected outcome was controlling action selection. OFC encoding of action selection prior to action initiation has been shown- and has been shown to be functionally relevant. Bradfield et al., 2015 nicely showed that OFC was functionally necessary to retrieve action consequence information and use that information to guide behavior where the action consequences were not visible but had to be inferred from recent experience. Perhaps the authors need to reframe the underlying novelty their analysis brings to the table.

Thanks for these comments. We fully agree with you that the result that OFC is involved in action initiation and selection is not novel. We have now stressed in the Introduction and Discussion (extensively rewritten to clarify all these important points) sections the parts of our work that are most novel, while acknowledging with higher precision previous work that the reviewer has mentioned:

1. Choice-related neuronal signals appear even before stimulus onset (this result has not been shown previously, except in a previous monkey report, quoted in the manuscript, by Padoa-Schioppa)
2. OFC represents the integration of ambiguous sensory information with prior information (our work develops a novel paradigm that studies the effect of past information in perceptual decision-making, characterized in detail by their effects on the psychometric curves)
3. OFC can represent prior information exclusively from the previous trial. This means that the representation of the state of OFC has ‘high temporal resolution’.

5. One major concern is the limited number of neurons sampled across only 3 rats. Neuron firing is assessed across only 3 rats; with Further, only a small number of neurons were sampled in each animal. That number was reduced almost in half when looking at neurons that showed significant responses, and the final number of neurons in each animal that showed changes was not reported.

Further, since firing was assessed and collapsed across many trials, there is a concern that the same neurons were continuously sampled and this sampling could skew the proportional data as well as bias response data. What was done to minimize this during data collection, or would the addition of rats to the analyses strengthen the data?

Thanks a lot for this important question. We are very happy to provide additional details and additional analysis (power analysis) to further support the statistical validity of our results.

First, a new power analysis (see below) shows that we have recorded enough neurons and animals to support our claims, given the strong statistical results that we have found ($p = 4.6 \times 10^{-9}$ in the central result of the manuscript). These results are extremely unlikely to change when increasing the sample size. It is important to note that our data sets were based on very long recordings per neuron, as the reviewer correctly points out. As these recordings were based on neurons that were independently sampled across sessions, long recordings, if anything, should provide much stronger statistical power to run any of the tests that we have used throughout our manuscript (note that we do not collapse neuronal responses across trials per neuron for our GLM analysis, as our analysis always uses all trials individually and tries to find correlations between regressors and responses on a trial-by-trial basis; furthermore, all neurons are considered on an equal footing, regardless of the number of trials recorded for each, such that we do not introduce biases in favor of neurons that have been recorded longer).

Following the reviewer’s advice, we now also report the number of neurons recorded for each animal after excluding irresponsive cells in the Supplementary Information.

Power Analysis. To address whether the sample size used in this study was sufficiently large, we performed a power analysis. The power (π) in hypothesis testing is

defined as the probability of correctly rejecting the null hypothesis (when it is false). It is related to the β , the probability of a type II error, through the equality $\pi = 1 - \beta$.

We first calculated the statistical power of the fraction of neurons that had their firing rate significantly modulated by each one of the reported regressors. Because significance for each neuron and regressor was calculated using a Binomial test (one-tailed), the power analysis was calculated using a Binomial distribution as well. First, we calculated the threshold on the fraction of neurons that would make us reject the null hypothesis when it is true. This value was the smallest fraction of neurons that fulfilled $p \leq 0.05$ for a one-tailed Binomial test. As this quantity only depends on the total number of neurons used in the Binomial test, it is the same across all regressors. We calculated the statistical power of the fraction of neurons encoding for a particular regressor as the probability of sampling a largest or equal value than the threshold from a binomial distribution, with probability of success defined as the actual fraction of neurons reported in the study. For a fixed probability of type I error ($\alpha = 0.05$) the statistical power increases with the strength of the result itself (reported fraction of neurons) and the number of samples used in the Binomial test (number of neurons). In other words, the probability of making a type II error when inferring the significance of that result decreases if these quantities increase.

The same procedure was applied to the statistical power analysis associated with the number of rats. Every rat that showed results in favor of the hypothesis was considered a success, and otherwise it was a failure. Because all rats in the study belonged to the success class (our main results are consistent across rats) we calculated the power if an additional rat without significant results (belonging to failure class) was added to the study.

We considered a result had sufficient statistical power when it exceeded the standard threshold (Baussell Barker, R. and Yu-Fang Li, 2002) $\pi = 0.80$. For the number of single units ($N=76$), the statistical power of the fraction of neurons encoding C_0 before stimulus presentation was $\pi=0.9996$, and $\pi=0.9947$ for X_{-1} . Thus, the two central results of our study are statistically rather solid.

Regarding the number of rats, we found significant results consistently across all rats. If we included a fourth rat with negative results the power of our conclusions would be $\pi = 0.95$. Thus, despite the small sample size, the strength of the results supported sufficient statistical power.

References

Baussell Barker, R., and Yu-Fang, Li. Power Analysis for Experimental Research: A Practical Guide for the Biological, Medical and Social Sciences. Cambridge University Press (2002).

REVIEWERS' COMMENTS:

Reviewer #1 (Remarks to the Author):

I appreciate the authors' thoughtful replies to my concerns. As is clear from all reviewers' comments, the OFC literature is a vast and tightly overlapping welter in which the process of decision making and decision formation has been attacked from almost every conceivable angle. As such, it is difficult for any one study to be dispositive. While I agree that this study does add to that body of knowledge, I would be more enthusiastic had it been able to clear a broader path through it. Nevertheless, it is a sound study, and if the authors have worked hard in this revised draft to place its results within the context of the field at large.

Reviewer #2 (Remarks to the Author):

The authors have done a great job addressing all of my concerns. I'll admit to being slightly surprised by the results of the power analysis, but the methods look correct so I'll accept the result!

I congratulate them on a very nice paper!

Reviewer #3 (Remarks to the Author):

The authors have adequately addressed my concerns. The addition of passive phase data strengthens their conclusions and raises some interesting hypotheses that are well addressed in the newly revised discussion.

Reviewers' comments

Reviewer #1

"I appreciate the authors' thoughtful replies to my concerns. As is clear from all reviewers' comments, the OFC literature is a vast and tightly overlapping welter in which the process of decision making and decision formation has been attacked from almost every conceivable angle. As such, it is difficult for any one study to be dispositive. While I agree that this study does add to that body of knowledge, I would be more enthusiastic had it been able to clear a broader path through it. Nevertheless, it is a sound study, and if the authors have worked hard in this revised draft to place its results within the context of the field at large."

We want to thank reviewer's comments on our manuscript. We have tried to provide the community with further evidence to understand the exact role of the orbitofrontal cortex in action selection, and we are glad to read that she/he appreciates our work.

Reviewer #2

"The authors have done a great job addressing all of my concerns. I'll admit to being slightly surprised by the results of the power analysis, but the methods look correct so I'll accept the result!"

I congratulate them on a very nice paper!"

We really thank the reviewer's comments on our manuscript. We found his comments very good and we wanted to provide him with the best answer possible.

Reviewer #3

"The authors have adequately addressed my concerns. The addition of passive phase data strengthens their conclusions and raises some interesting hypotheses that are well addressed in the newly revised discussion."

We want to thank the reviewer for her/his words. We also think the addition of passive data further strengthen our hypothesis and opens the possibility for further studies.